# PBK/TOPK mediates Ikaros, Aiolos and CTCF displacement from mitotic chromosomes and alters chromatin accessibility at selected C2H2-zinc finger protein binding sites

Andrew Dimond [1,2] ✉, Do Hyeon Gim [1], Elizabeth Ing-Simmons [3], Chad Whilding[4], Holger B. Kramer[5], Dounia Djeghloul [1,2], Alex Montoya [6], Bhavik Patel[7], Sherry Cheriyamkunnel[1,2], Karen E. Brown[1,2], Pavel V. Shliaha [6], Juan M. Vaquerizas [3], Matthias Merkenschlager [8] & Amanda G. Fisher [1,2] ✉

PBK/TOPK is a mitotic kinase implicated in haematological and non-haematological cancers. Here we show that the key haemopoietic regulators Ikaros and Aiolos require PBK-mediated phosphorylation to dissociate from chromosomes in mitosis. Eviction of Ikaros is rapidly reversed by addition of the PBK-inhibitor OTS514, revealing dynamic regulation by kinase and phosphatase activities. To identify more PBK targets, we analysed loss of mitotic phosphorylation events in *Pbk*[−/−] preB cells and performed proteomic comparisons on isolated mitotic chromosomes. Among a large pool of C2H2-zinc finger targets, PBK is essential for evicting the CCCTC-binding protein CTCF and zinc finger proteins encoded by *Ikzf1*, *Ikzf3*, *Znf131* and *Zbtb11*. PBK-deficient cells were able to divide but showed altered chromatin accessibility and nucleosome positioning consistent with CTCF retention. Our studies reveal that PBK controls the dissociation of selected factors from condensing mitotic chromosomes and contributes to their compaction.

To improve our understanding of how cell identity is maintained when cells divide, there has been renewed interest in discriminating the factors that either remain associated with mitotic chromosomes or are displaced as chromosomes condense towards metaphase[1–14]. This discrimination is not entirely straightforward, as fixation of mitotic samples has been shown to artificially strip some chromatin-associated factors from mitotic chromosomes[15]. In addition, although it is known that phosphorylation, mediated by mitotic kinases, can drive dissociation of proteins from mitotic chromosomes[16–24] there is a paucity of knowledge regarding the extent to which factors are actively displaced, and which kinases, either alone or in combination, provoke the eviction of specific DNA-binding or chromatin-associated proteins. To close this knowledge gap, we have examined changes in the cell cycle distribution of key factors that regulate and determine cell commitment. Ikaros is a much studied lymphoid-restricted C2H2-zinc finger (ZF) transcription factor, encoded by the *Ikzf1* gene, that is

[1]Epigenetic Memory Group, MRC LMS, Imperial College London, Hammersmith Hospital Campus, London, UK. [2]Department of Biochemistry, University of Oxford, Oxford, UK. [3]Developmental Epigenomics Group, MRC LMS, Imperial College London, Hammersmith Hospital Campus, London, UK. [4]Microscopy Facility, MRC LMS, Imperial College London, Hammersmith Hospital Campus, London, UK. [5]Mass Spectrometry Facility, MRC LMB, Francis Crick Avenue, Cambridge Biomedical Campus, Cambridge, UK. [6]Proteomics and Metabolomics Facility, MRC LMS, Imperial College London, Hammersmith Hospital Campus, London, UK. [7]Flow Cytometry Facility, MRC LMS, Imperial College London, Hammersmith Hospital Campus, London, UK. [8]Lymphocyte Development Group, MRC LMS, Imperial College London, Hammersmith Hospital Campus, London, UK. ✉e-mail: andrew.dimond@bioch.ox.ac.uk; amanda.fisher@bioch.ox.ac.uk

critical for lymphocyte specification and differentiation, and has regulatory roles throughout the development of both B and T-cell lineages[25–32]. At the molecular level, Ikaros forms dimers with itself or other close family members (such as Aiolos)[30,33–35], and is postulated to either mediate silencing or to activate transcription, according to context[36–40]. Ikaros proteins have also been shown to interact with the nucleosome remodelling and deacetylation complex (NuRD) in B and T-cell lineages[40–45], and to bind at pericentric heterochromatin domains in lymphocytes during interphase[35,46–48].

In a previous study[16] Ikaros was shown to be inactivated in mitosis, potentially as part of a common mechanism shared with other proteins that contain C2H2-ZF DNA binding domains[16,17,49,50]. At the time, this result was considered surprising because Ikaros is implicated in the heritable silencing of gene activity[47] and could therefore be presumed to remain at sites of repression. Furthermore, prior studies by us and others have shown that many factors that contribute to durable chromatin-based epigenetic silencing remain chromosome-associated in mitosis[8,10,51–53]. This includes components of the DNA methylation machinery DNMT1,3A,3B; Polycomb Repressor Complexes 1 and 2; and the histone H3 lysine 9 methyltransferases SUV39H1/H2.

To verify that Ikaros does indeed dissociate from chromosomes during mitosis, and to better define the kinases responsible for this eviction, we have generated tools to enable real-time imaging of Ikaros distribution in living cells. We have used these to visualise Ikaros redistribution during the cell cycle and following treatment with a range of different kinase inhibitors. We find that PDZ-binding kinase (PBK, also known as T-lymphokine-activated killer-cell-originated protein kinase (TOPK)) is both necessary and sufficient to evict Ikaros from metaphase chromosomes, a finding confirmed by genetic knock-out (KO) of *Pbk*. Furthermore, treatment of Ikaros-depleted chromosomes with the PBK-inhibitor OTS514 provokes rapid re-association of Ikaros with condensed chromosomes, revealing dynamic regulation by kinase and phosphatase activities. To uncover other factors targeted for eviction by PBK/TOPK, we have adopted a two-pronged approach. Firstly, we have raised an antibody to a phosphorylated C2H2-ZF peptide linker and performed immunoprecipitation (IP) from mitotic lysates of B cell progenitors (preB cells) to screen for potential targets. Secondly, using a previously described approach[10,53], we have purified unfixed mitotic chromosomes from *Pbk*[−/−] or wildtype (WT, *Pbk*[+/+]) preB lymphocytes and compared their proteomes by quantitative liquid chromatography-tandem mass spectrometry (LC-MS/MS). This combined strategy, together with immunofluorescence (IF) comparisons, enables us to identify and verify an important cohort of factors that are coordinately evicted from chromosomes by PBK. These include Ikaros, Aiolos, the DNA-binding factors ZNF131, ZBTB11 and BCL11A, and the insulator protein CTCF. Chromatin accessibility data (using ATAC-seq) are consistent with greater retention of C2H2-ZF factors on *Pbk*[−/−] mitotic chromosomes, and we observe increased accessibility at multiple loci, especially at CTCF binding sites. *Pbk*[−/−] metaphase chromosomes are significantly less condensed that their normal counterparts, a result which suggests that PBK-mediated phosphorylation, and the subsequent displacement of C2H2-ZF factors, may be important to enable chromosomes to properly condense in mitosis.

## Results

### Ikaros is actively evicted from mitotic chromosomes by phosphorylation

Previous work has suggested that Ikaros binds to DNA in a cell-cycle-dependent manner[16,46,54]. Antibody-based staining of mouse preB cells confirmed that Ikaros localises to pericentric (heterochromatin) clusters in interphase, but is absent from mitotic chromosomes during metaphase, with association being re-established during mitotic exit (Fig. 1a). This cell-cycle-dependent localisation of Ikaros was also seen in mouse T (VL3-3M2) and macrophage (J774A.1) cell lines (Supplementary Fig. 1a, b) and is consistent with previous reports[46]. To

independently verify that the mitotic eviction of Ikaros was not an artefact of sample fixation, we used CRISPR/Cas9 to engineer knock-in (KI) mouse preB cells that express Ikaros-mNeonGreen fusion proteins derived from the endogenous *Ikzf1* locus (Fig. 1b). Two different single guide RNAs (sgRNAs) were used to generate four hetero- or homozygous KI clones expressing Ikaros-mNeonGreen fusion proteins (Fig. 1b, middle/lower). Live-cell imaging of these clones revealed that Ikaros-mNeonGreen dissociates from condensing chromosomes during prophase and is no longer detected on chromosomes at metaphase (Fig. 1c, Supplementary Fig. 1c and Supplementary Movies 1–4). Ikaros-mNeonGreen showed re-association with pericentric chromosomal regions during anaphase, and at telophase/cytokinesis. These results importantly validate our antibody-based observations and reveal that Ikaros re-association with chromosomes occurs rapidly towards the end of mitosis, prior to entry into G1 (Supplementary Movies 1–4).

Previous literature suggests that phosphorylation can regulate the eviction of many DNA-binding factors in mitosis[16–24]. To better understand this, and to identify the mechanisms of Ikaros dissociation and re-association, actively dividing *Ikzf1*[mNG/mNG] preB cells were treated with a range of kinase inhibitors (Fig. 1d and Supplementary Fig. 1d; as described in Methods and listed in Supplementary Table 3). By examining living cells during metaphase, we found that three inhibitors prevented Ikaros-mNeonGreen eviction from metaphase chromosomes, compared with DMSO-treated controls: K252a (a broad-spectrum kinase inhibitor; Fig. 1d), and OTS514 and OTS964 (two related PBK inhibitors; Fig. 1d and Supplementary Fig. 1d). Cells treated with each of these inhibitors showed abundant Ikaros signal on metaphase chromosomes, particularly focused around centromeric domains. In contrast, treatment with other inhibitors, including those directed towards Aurora kinases (VX-680, Hesperadin), cyclin dependent kinases (Alsterpaullone (Alp)) or the known Ikaros kinase Casein kinase 2 (CX-4945)[55], produced no appreciable change in Ikaros-mNeonGreen distribution as compared with controls (Fig. 1d, left and Supplementary Fig. 1d). Quantification of DNA/cytoplasmic signal ratios (Supplementary Fig. 1e) confirmed that significant changes in Ikaros localisation were only seen following treatment with broad or PBK-specific inhibitors (Fig. 1d, right). The capacity of OTS514 to block Ikaros eviction from metaphase chromosomes was verified using a second Ikaros-mNeonGreen KI clone (Supplementary Fig. 1f), as well as by anti-Ikaros staining of normal preB cells expressing WT Ikaros (Fig. 1e). Treatment of T cells and a macrophage cell line with OTS514 resulted in a similar retention of Ikaros on metaphase chromosomes (Fig. 1f, g), consistent with a common mechanism of phosphorylation-induced mitotic eviction in different cell types.

### Ikaros distribution is dynamically regulated by competing PBK and phosphatase activities

Ikaros binding to chromosomes during the cell cycle appears highly dynamic, with rapid dissociation and re-association occurring during normal mitotic progression (Supplementary Movies 1–4). To determine whether the normal dissociation of Ikaros from chromosomes at metaphase could be reversed (rather than simply prevented) by PBK inhibition, preB cells were arrested with demecolcine (allowing Ikaros-mNeonGreen to dissociate) before treating with OTS514 (Supplementary Fig. 2a). Ikaros re-association with chromosomes was observed exclusively following inhibitor treatment, suggesting that mitotic eviction can be reversed upon loss of PBK activity. In order to observe this reversal directly, individual mitotic cells in which Ikaros-mNeonGreen had already become dissociated were treated and monitored by live-cell imaging (Fig. 2a). Immediately following the addition of OTS514 inhibitor, an increased signal was observed at the centromeric regions of chromosomes, which rapidly increased over a period of minutes (Fig. 2a, quantified right). Given the speed of this re-association, we hypothesised that there might be an active mechanism to reverse Ikaros phosphorylation. In support of this, we showed that

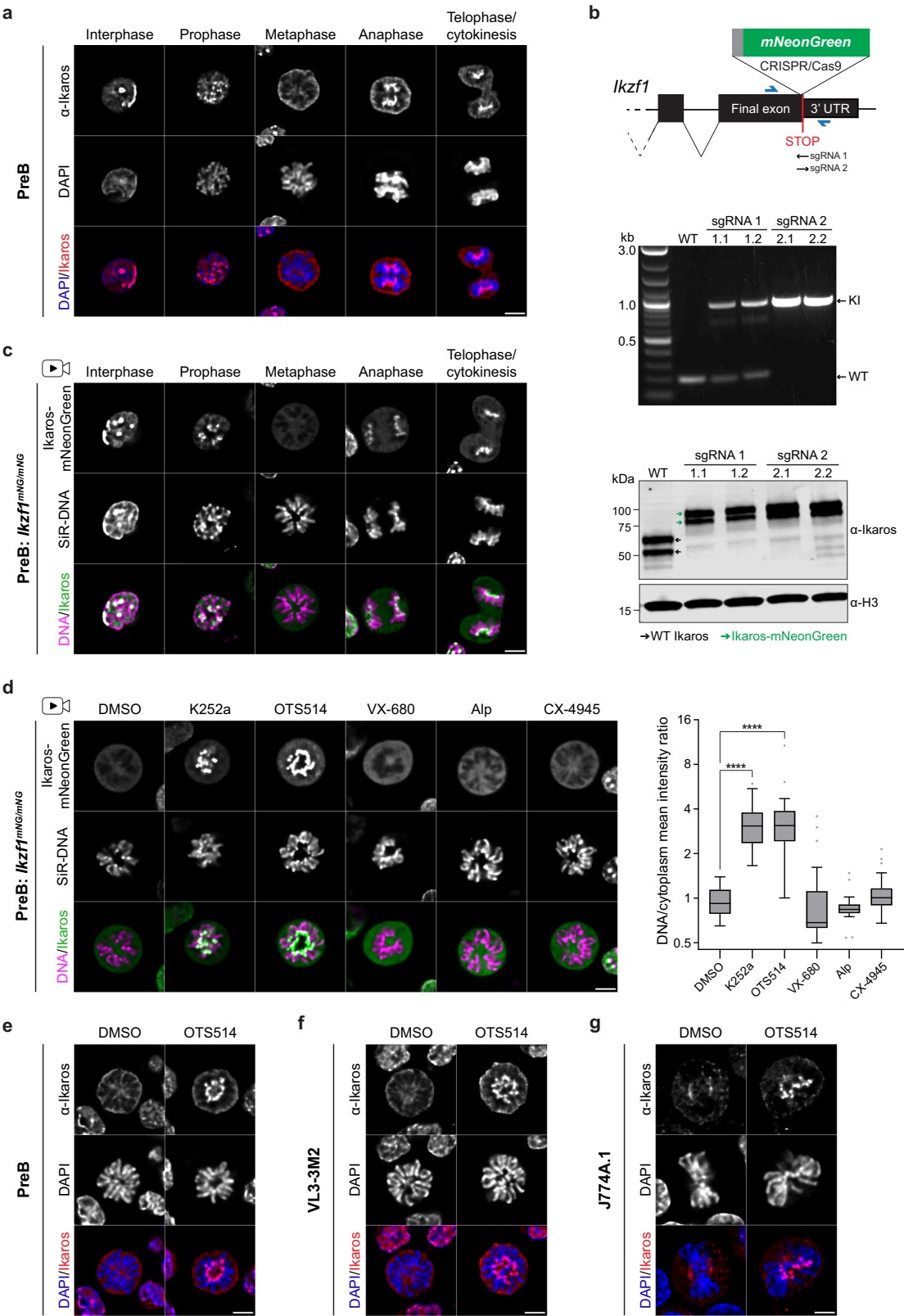

co-treatment with Calyculin A, a potent PP1 and PP2A inhibitor, blocked OTS514-induced Ikaros re-association with mitotic chromosomes (Fig. 2b). This result was also verified by antibody staining of Ikaros in WT preB cells (Supplementary Fig. 2b). In contrast, Okadaic acid (OA), a PP2A phosphatase inhibitor with a lower potency for PP1, did not prevent PBK-inhibitor-driven re-association of Ikaros (Fig. 2b), suggesting PP1 may be responsible for Ikaros dephosphorylation.

Although OTS514 and OTS964 are reported to be inhibitors of PBK, both can exhibit some off-target activity against other kinases[56,57]. Therefore, to confirm that PBK (a kinase reported to be selectively active in prophase and metaphase[50,58]) regulates the mitotic eviction of Ikaros from chromosomes, we undertook a genetic approach. PBK-KO ($Pbk^{-/-}$) lines were generated from WT and $Ikzf1^{mNG/mNG}$ mouse preB cells using CRISPR/Cas9 (see Methods for details). PBK-null cells were

**Fig. 1 | Ikaros dissociates from metaphase chromosomes but can be retained following treatment with specific kinase inhibitors. a** Ikaros staining in fixed, asynchronous mouse preB cells, showing representative interphase and mitotic stages. Scale bar = 5 μm; ≥12 cells imaged per stage (> 50 metaphase) across four independent experiments. **b** CRISPR/Cas9-mediated knock-in (KI) of mNeonGreen at endogenous *Ikzf1* in mouse preB cells (illustrated top). Primers (blue arrows) were used to confirm KI by PCR (middle, two clones per targeting). Sequencing revealed clones 1.1 and 1.2 are heterozygous (*Ikzf1^{mNG/-}*, apparent WT alleles harbour frameshift mutations); clones 2.1 and 2.2 are homozygous (*Ikzf1^{mNG/mNG}*). Immunoblotting (bottom) detects Ikaros-mNeonGreen (anti-Ikaros(C-terminal), 1:5000); anti-H3 = loading control. **c** Live-cell imaging of Ikaros-mNeonGreen in asynchronous *Ikzf1^{mNG/mNG}* mouse preB cells (clone 2.1) cultured with SiR-DNA, showing representative interphase and mitotic stages. Scale bar = 5 μm; ≥11 cells imaged per stage (> 50 metaphase) across four independent experiments. **d** Live-cell imaging of mitotic *Ikzf1^{mNG/mNG}* mouse preB cells (clone 2.1) from asynchronous cultures treated 10 min with DMSO or kinase inhibitors (K252a 1 μM, others 10 μM; Alp = alsterpaullone). Representative of at least two independent experiments (five for K252a and OTS514); scale bar = 5 μm. Chromosomal versus cytoplasmic

mNeonGreen intensity is quantified (illustrated in Supplementary Fig. 1e) for a representative replicate of each treatment (n = 20, 27, 25, 18, 28, 26 cells/treatment; boxplots show median, interquartile range, Tukey whiskers (log2 scale); ****$p_{adj}$ < 0.0001, two-tailed Dunnett's test). **e** Ikaros staining in mitotic preB cells (fixed asynchronous cultures) following 10 min DMSO/OTS514 (10 μM) treatment. Representative of three independent experiments. All (63/63) OTS514-treated metaphase cells showed bright centromeric staining, versus 0/66 for DMSO. Scale bar = 5 μm. **f** Ikaros staining in mitotic VL3-3M2 mouse T cells (fixed asynchronous cultures) following 10 min DMSO/OTS514 (10 μM) treatment. Representative of three independent experiments; 79% of OTS514-treated metaphase cells showed centromeric staining (n = 80 cells; 34 = bright, 29 = mid, 17 = low/none), versus 17% of DMSO-treated cells (n = 75 cells; 0 = bright, 13 = mid, 62 = low/none). Scale bar = 5 μm. **g** Ikaros staining in mitotic J774A.1 mouse macrophages (fixed asynchronous cultures) after 10 min DMSO/OTS514 (10 μM) treatment. Representative of two independent experiments. All (42/42) OTS514-treated metaphase cells showed bright centromeric staining, versus 0/46 for DMSO. Scale bar = 5 μm. Source data for Fig. 1b, d are provided as a Source Data file.

capable of division and expressed Ikaros protein at levels comparable to the parental WT (Fig. 2c). During interphase, *Pbk^{-/-}* cells displayed a normal pattern of pericentric Ikaros staining (Supplementary Fig. 2c). However, in mitosis we observed abnormal retention of Ikaros at metaphase chromosomes, with bright centromeric labelling in all mitotic *Pbk^{-/-}* cells (Fig. 2d). By targeting *Pbk* in preB KI cells expressing Ikaros-mNeonGreen (Fig. 2e) we were also able to evaluate the lack of PBK in live cells. *Ikzf1^{mNG/mNG} Pbk^{+/+}* and *Pbk^{-/-}* cells expressed comparable levels of Ikaros-mNeonGreen fusion protein (Fig. 2e, lower), with signal localising to pericentric heterochromatin clusters in interphase, regardless of PBK status (Supplementary Fig. 2d). However, as cells entered mitosis, Ikaros-mNeonGreen was seen to remain chromosome-associated in the absence of PBK (Fig. 2f, quantified right). Interestingly, although treatment with Calyculin A was able to block the OTS514-indcued re-association of Ikaros (Fig. 2b), it was insufficient to evict bound Ikaros in the absence of PBK (Supplementary Fig. 2e). Together, these results show that Ikaros dissociation and re-association with mitotic chromosomes is dynamically and actively regulated through opposing kinase (PBK) and phosphatase (PP1) activities.

## PBK regulates mitotic chromosome compaction and protein composition

To determine whether PBK loss has wider impacts on mitotic chromosomes, beyond Ikaros retention, we used flow cytometry to purify native mitotic chromosomes[10,53] from *Pbk^{+/+}* and *Pbk^{-/-}* mouse preB cells for downstream analysis (illustrated in Supplementary Fig. 3a). We isolated two representative individual chromosomes, 3 and 19, from *Pbk^{+/+}* and *Pbk^{-/-}* cells (Supplementary Fig. 3b and Fig. 3a) and found that chromosomes (and centromeres) from PBK KO cells were significantly larger than equivalents isolated from WT cells (Fig. 3b, quantified in Fig. 3c). This increase in mitotic chromosome size in *Pbk^{-/-}* cells was additionally verified using conventional metaphase spreads (Fig. 3d and Supplementary Fig. 3c) and shows that PBK activity is required for the normal compaction of mitotic chromosomes.

PBK has previously been implicated in regulating multiple targets in mitosis, particularly other C2H2-ZF transcription factors[50]. To investigate whether loss of PBK affects retention of other C2H2-ZF factors by mitotic chromosomes, or more broadly other chromatin components, we isolated total mitotic chromosomes from *Pbk^{+/+}* and *Pbk^{-/-}* mouse preB cells (Supplementary Fig. 3d), and subjected these to proteomics analysis, as previously described[10,53] (and detailed in Methods). To determine the factors enriched on mitotic chromosomes in each condition, we performed LC-MS/MS analysis of sorted chromosomes, compared to pre-sorted total mitotic lysate pellets (illustrated in Supplementary Fig. 3a and Supplementary Data 1).

Replicates were similar for both *Pbk^{+/+}* and *Pbk^{-/-}* conditions, whilst sorted and unsorted samples clustered separately (Supplementary Fig. 3e). Similar numbers of proteins were detected in WT and KO conditions, with a comparable proportion showing significant enrichment (17.1% versus 17.8%) or depletion (25.2% versus 29.9%) from mitotic chromosomes (Supplementary Fig. 3f).

As expected, both *Pbk^{+/+}* and *Pbk^{-/-}* mitotic chromosomes showed an enrichment for histone proteins (Fig. 3e, top left) and condensin complex proteins (Fig. 3e, middle left), while components of the proteasome were not significantly enriched (Fig. 3e, top right). Amongst cohesin complex components, only Shugoshin proteins (Sgo1/2) were significantly enriched on *Pbk^{+/+}* and *Pbk^{-/-}* chromosomes (Fig. 3e, middle right), consistent with the general eviction of cohesin during prophase (although depletion was less pronounced in *Pbk^{-/-}* samples), and the selective protection of centromeric cohesin by Shugoshin until anaphase[59]. Inspection of C2H2-ZF factors revealed that most, including Ikaros (Ikzf1) and Aiolos (Ikzf3), were either depleted or showed no significant enrichment on mitotic chromosome samples derived from WT (*Pbk^{+/+}*) cells (Fig. 3e, lower left). In *Pbk^{-/-}* samples, we detected more C2H2-ZF factors overall (52 versus 43) and the proportion showing co-enrichment with chromosomes nearly tripled. Ikaros and Aiolos were among the factors enriched exclusively on *Pbk^{-/-}* chromosomes, whilst other factors, such as BCL11A, no longer showed significant depletion in the absence of PBK. Despite an increase in mitotic chromosome retention of multiple C2H2-ZF factors in *Pbk^{-/-}* samples, approximately two-thirds remained as not significantly enriched. These results suggest that PBK activity is required for the dissociation of a subset of C2H2-ZF proteins from mitotic chromosomes, whilst others can be evicted by alternative mechanisms.

To further examine changes in the association of factors with *Pbk^{-/-}* mitotic chromosomes, we performed gene ontology (GO) analysis of depleted or enriched factors (Supplementary Fig. 3g). Among the factors defined as depleted from *Pbk^{+/+}* and *Pbk^{-/-}* chromosomes, there was a broad similarity in overrepresented terms (Supplementary Fig. 3g, lower), with spliceosomal components showing mitotic depletion in both conditions. However, among the proteins defined as enriched, we observed differences in overrepresented terms between *Pbk^{+/+}* and *Pbk^{-/-}* samples (Supplementary Fig. 3g, upper). This included a modest reduction in outer-kinetochore proteins, a small increase in heterochromatin-associated proteins, and an apparent enrichment of nucleosome remodelling complexes, including ISWI-, SWI/SNF superfamily-, INO80- and CHD-type complexes, and exemplified by the NuRD complex (Fig. 3e, lower right). Taken together, these results indicate that PBK activity is required for the mitotic eviction of a subset of C2H2-ZF proteins, may affect the normal dissociation of nucleosome remodelling complexes, and is important for normal chromosome compaction.

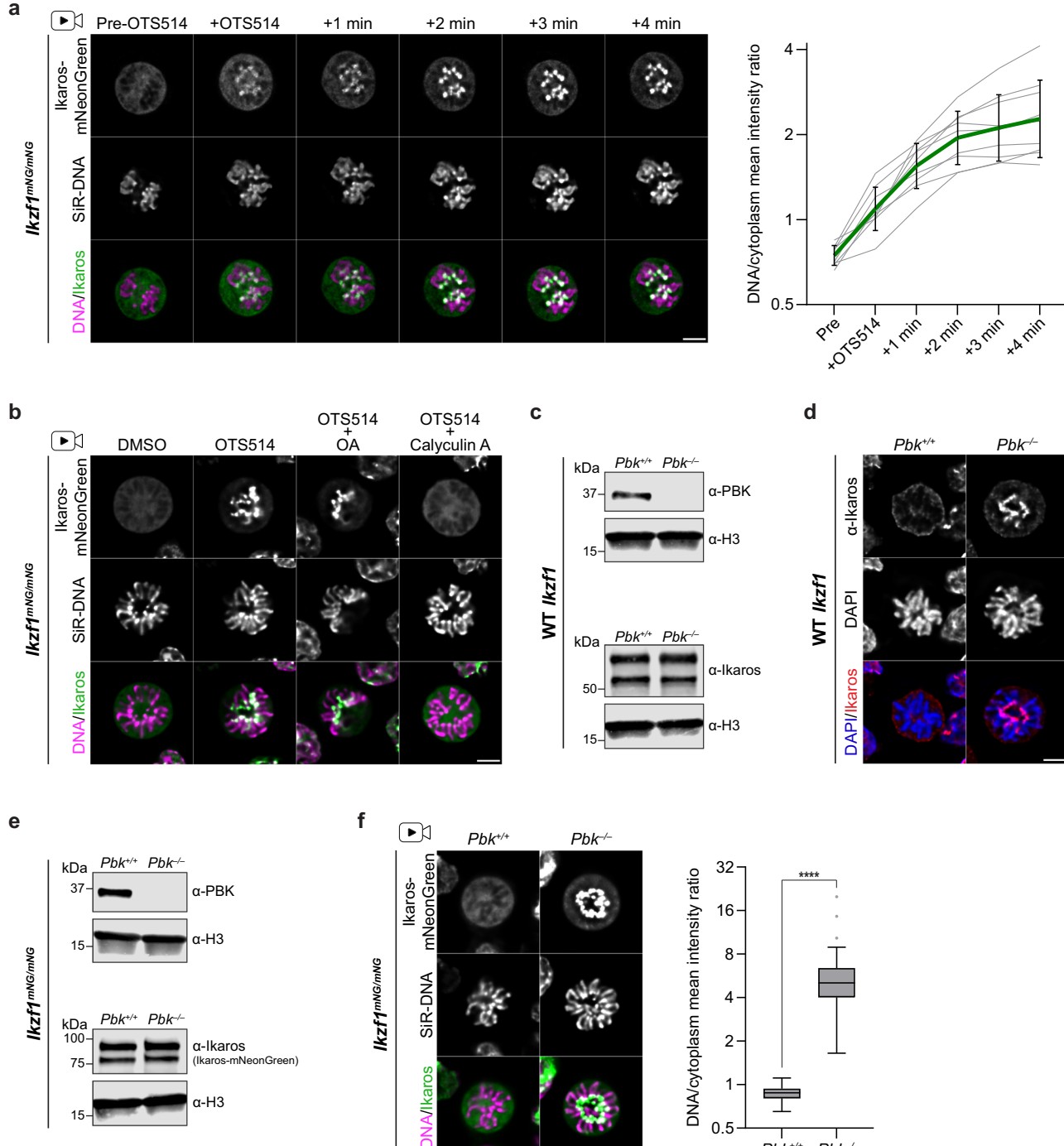

## PBK directly phosphorylates multiple C2H2-ZF proteins and regulates mitotic dissociation of CTCF

To further investigate potential targets of PBK, we generated a specific anti-phospho-linker antibody. Ikaros has previously been shown to be phosphorylated at specific sites within the highly conserved C2H2-ZF linkers[16,54], and we therefore used a phosphorylated Ikaros136-143 linker peptide to generate rabbit antisera (as illustrated in Fig. 4a, see Supplementary Table 2 for details). This antibody detected an array of proteins in mitotic lysates from WT preB cells which were only weakly detectable in asynchronous lysates, and which were not detected in *Pbk⁻/⁻* samples (Fig. 4b). Enhanced phosphorylation of histone H3 at serine 10 (H3S10p) was detected in mitotic versus asynchronous samples as expected, but levels were comparable in *Pbk⁻/⁻* and *Pbk⁺/⁺* samples, despite previous suggestions that PBK may contribute to H3S10

phosphorylation[60]. To identify proteins that were recognised by the anti-phospho-linker antibody, we performed quantitative proteomic comparisons of *Pbk⁺/⁺* and *Pbk⁻/⁻* mitotic lysates following immunoprecipitation (IP; Supplementary Data 2). Although protein abundances in total mitotic lysates appeared similar between conditions (Supplementary Fig. 4a), we detected a wide range of factors in WT samples following IP with the anti-phospho-linker antibody, which were absent or significantly reduced following IP from *Pbk⁻/⁻* samples (Fig. 4c, high/low values are shown by a blue/yellow scale). Overall, 111 hits were detected solely, or significantly more abundantly, in precipitates derived from *Pbk⁺/⁺* lysates, as compared to *Pbk⁻/⁻* lysates. Whilst unlikely to be an exhaustive list of substrates, this provides a catalogue of putative mitotic phosphorylation targets of PBK. The majority of these factors (72%) correspond to C2H2-ZF proteins (highlighted in red,

**Fig. 2 | Ikaros dissociation is regulated by competing PBK and phosphatase activities. a** Live-cell imaging of Ikaros-mNeonGreen re-localisation in individual mitotic *Ikzf1*<sup>mNG/mNG</sup> mouse preB cells before and after 10 μM OTS514 addition (clone 2.1, asynchronous cultures pre-incubated with SiR-DNA). Shown is a representative cell (scale bar = 5 μm) and quantification of mean chromosomal/cytoplasmic mNeonGreen signal for nine cells from six independent experiments (grey = individual cells, green = geometric mean with geometric standard deviation, log2 scale). **b** Live-cell imaging of mitotic *Ikzf1*<sup>mNG/mNG</sup> mouse preB cells (clone 2.1) from asynchronous cultures (pre-cultured with SiR-DNA) following 10 min treatment with DMSO or 10 μM OTS514 alone or in combination with Calyculin A (100 nM) or Okadaic Acid (OA, 1 μM). Representative of three independent replicates; scale bar = 5 μm. Bright centromeric signal was observed in all OTS514-treated (25/25) and OTS514 + OA-treated (28/28) metaphase cells, versus very weak/no signal in 96% of Calyculin A co-treated cells (20/27 = none, 6/27 = very weak). **c** Western blot of CRISPR/Cas9-generated *Pbk*<sup>−/−</sup> mouse preB cells confirms PBK loss (anti-PBK 1:1000) and shows similar Ikaros levels (anti-Ikaros(C-terminal) 1:1000) in *Pbk*<sup>+/+</sup> and *Pbk*<sup>−/−</sup> cells. Anti-H3 = histone H3 loading control. Representative of two

independent experiments; validated in a second *Pbk*<sup>−/−</sup> clone. **d** Ikaros staining in WT (*Pbk*<sup>+/+</sup>) or *Pbk*<sup>−/−</sup> mitotic preB cells (fixed asynchronous cultures). Representative of three independent experiments. All (30/30) *Pbk*<sup>−/−</sup> metaphase cells showed bright centromeric staining versus only 1/54 with a weak signal in *Pbk*<sup>+/+</sup>. Scale bar = 5 μm. Results were further validated in a second *Pbk*<sup>−/−</sup> clone. **e** Western blot validation of PBK KO in *Ikzf1*<sup>mNG/mNG</sup> (clone 2.1) mouse preB cells (anti-PBK 1:1000) and comparison of Ikaros-mNeonGreen protein levels (anti-Ikaros(C-terminal) 1:1000) in *Pbk*<sup>+/+</sup> and *Pbk*<sup>−/−</sup> cells. Anti-H3 = loading control. Representative of two independent experiments. **f** Live-cell imaging of *Pbk*<sup>+/+</sup> or *Pbk*<sup>−/−</sup> mitotic mouse *Ikzf1*<sup>mNG/mNG</sup> preB cells from asynchronous cultures incubated with SiR-DNA. Images are representative of >70 mitotic cells from three independent experiments. Scale bar = 5 μm. Mean chromosomal/cytoplasmic mNeonGreen intensity is quantified for one representative replicate (*Pbk*<sup>+/+</sup> *n* = 32 cells, *Pbk*<sup>−/−</sup> *n* = 34 cells; boxplots show median, interquartile range and Tukey whiskers (log2 scale); ****$p < 0.0001$, unpaired two-tailed *t* test). Source data for Fig. 2a, c, e, f are provided as a Source Data file.

Fig. 4c), including SP1, YY1, CTCF, Ikaros and Aiolos. The differential detection of phosphorylated proteins between *Pbk*<sup>−/−</sup> and *Pbk*<sup>+/+</sup> samples was not caused by an underlying change in cell-cycle distribution or a reduced efficiency of mitotic arrest in mutant cells (Supplementary Fig. 4b). Furthermore, since acute treatment of mitotically arrested WT preB cells with OTS514 resulted in a loss of antibody-reactive proteins (Supplementary Fig. 4c), these results are consistent with a requirement for PBK to mediate and maintain phosphorylation of a cohort of proteins in mitosis.

We next compared PBK targets identified by anti-phospho-linker IP with those identified by proteomic analysis of flow-sorted mitotic chromosomes. While there was only a modest overlap in these datasets, more C2H2-ZF PBK targets were detected overall in *Pbk*<sup>−/−</sup> chromosome samples than in *Pbk*<sup>+/+</sup> samples (Supplementary Fig. 4d), consistent with a greater retention of these factors in the absence of PBK activity. It was noticeable however that despite a loss of phosphorylation, there was not evidence of a wholescale redistribution of C2H2-ZF phosphorylation targets in the absence of PBK, and most of these factors were not significantly enriched on *Pbk*<sup>−/−</sup> chromosomes (Supplementary Fig. 4d). Instead, a small subset of PBK phosphorylation targets, namely Ikaros, Aiolos, ZNF131 and ZBTB11, showed a significant enrichment exclusively on *Pbk*<sup>−/−</sup> chromosomes, while BCL11A was no longer depleted.

Immunofluorescence analysis of selected C2H2-ZF factors in *Pbk*<sup>+/+</sup> and *Pbk*<sup>−/−</sup> cells confirmed that, whereas Ikaros requires PBK for mitotic dissociation, release of SP1 and YY1 is independent of PBK phosphorylation (Fig. 4d). In contrast, CTCF showed increased retention on the arms of metaphase *Pbk*<sup>−/−</sup> chromosomes (Fig. 4d), whereas in WT *Pbk*<sup>+/+</sup> preB cells, CTCF is largely dissociated. Surprisingly, CTCF had not been previously identified as significantly co-enriched on *Pbk*<sup>−/−</sup> mitotic chromosomes by proteomics. This could reflect technical limitations in detection that stem from CTCF being in dynamic flux[10]. To rule out the possibility that demecolcine arrest had affected CTCF detection, we performed immunofluorescence on mitotically arrested *Pbk*<sup>−/−</sup> preB cells and confirmed that both CTCF and Ikaros remain chromosome-associated (Supplementary Fig. 4e). Differences in mitotic localisation of factors was not due to changes in protein levels between *Pbk*<sup>+/+</sup> and *Pbk*<sup>−/−</sup> cells (Supplementary Fig. 4f, g and Fig. 2c). Instead, we hypothesise that some PBK-independent factors might have additional sites of phosphorylation which are sufficient to evict these factors even in the absence of linker phosphorylation. In support of this, YY1 is reported to be phosphorylated by Aurora A at serine 365 (adjacent to the PBK linker phosphorylation site)[19], and we noticed that SP1 also contains a potential Aurora A target motif (Supplementary Fig. 4h). In contrast, PBK-dependent factors Ikaros, Aiolos, CTCF, ZNF131 and ZBTB11 all lack putative Aurora A target motifs. We further validated that mitotic eviction of CTCF and Ikaros, but not SP1 and YY1, is PBK-

dependent by acute OTS514 inhibitor treatment (Supplementary Fig. 4i). Interestingly, CTCF appeared to re-associate with slower kinetics than Ikaros following PBK inhibition. This is consistent with the trend during mitotic exit, in which CTCF remains dissociated until telophase/cytokinesis (Supplementary Fig. 4j), while Ikaros begins to re-associate in anaphase (Fig. 1a, b), suggestive of different dephosphorylation rates. Collectively, our data indicate that PBK-mediated phosphorylation is necessary and sufficient for the mitotic eviction of a subset of C2H2-ZF transcription factors that includes Ikaros, Aiolos, ZNF131, ZBTB11, BCL11A and CTCF.

## Loss of PBK alters mitotic chromatin accessibility at C2H2-ZF protein binding sites

We have shown that mitotic chromosomes isolated from *Pbk*<sup>−/−</sup> cells, preB cells were less compact (Fig. 3b–d) and more enriched for certain C2H2-ZF binding proteins (Fig. 3e, lower left; and Fig. 4d) than WT equivalents. To understand the impacts of PBK-mediated phosphorylation on chromatin accessibility, we performed ATAC-seq on isolated native mitotic chromosomes from *Pbk*<sup>+/+</sup> and *Pbk*<sup>−/−</sup> cells, that were purified by flow cytometry to ensure high mitotic purity irrespective of synchronisation efficiency (illustrated in Fig. 5a). Fragment size distributions, corresponding to both nucleosome-free and nucleosomal fragments, were very similar between conditions (Supplementary Fig. 5a). Overall, chromatin accessibility was also broadly similar, although *Pbk*<sup>+/+</sup> and *Pbk*<sup>−/−</sup> samples clearly segregated by principal component analysis (Supplementary Fig. 5b). Differential accessibility analysis confirmed that most peaks (~86%) were unchanged but revealed ~15,000 differentially accessible peaks (Fig. 5b), of which most (11,479 versus 3811) showed increased accessibility in the absence of PBK. Regions of increased accessibility are strongly biased towards transcription start sites (TSSs), compared to the distribution of peaks which are unchanged or lose accessibility (Fig. 5c). By examining individual genomic loci, we confirmed that accessibility was largely unchanged in the absence of PBK at most regions, including at *Ikzf1* (Supplementary Fig. 5c, upper) and, perhaps surprisingly, at known Ikaros target genes such as *Igll* and *Vpreb1*[36–38,44] (Supplementary Fig. 5c, lower). However, specific differences were evident at some loci (Fig. 5d and Supplementary Fig. 5d), including gains of accessibility at previously inaccessible regions and more modest alterations at peaks already present in WT samples. Many increased peaks were located at TSSs (Fig. 5c, d), which were enriched for genes involved in mRNA processing, translation, and protein degradation (Supplementary Fig. 5e, upper); whilst genes with decreased TSS accessibility showed minimal GO term enrichment (Supplementary Fig. 5e, lower). We also observed that many sites gaining ATAC-seq signal appeared to overlap with known CTCF binding sites (Fig. 5d). To further investigate factors which might be responsible for this differential accessibility, we

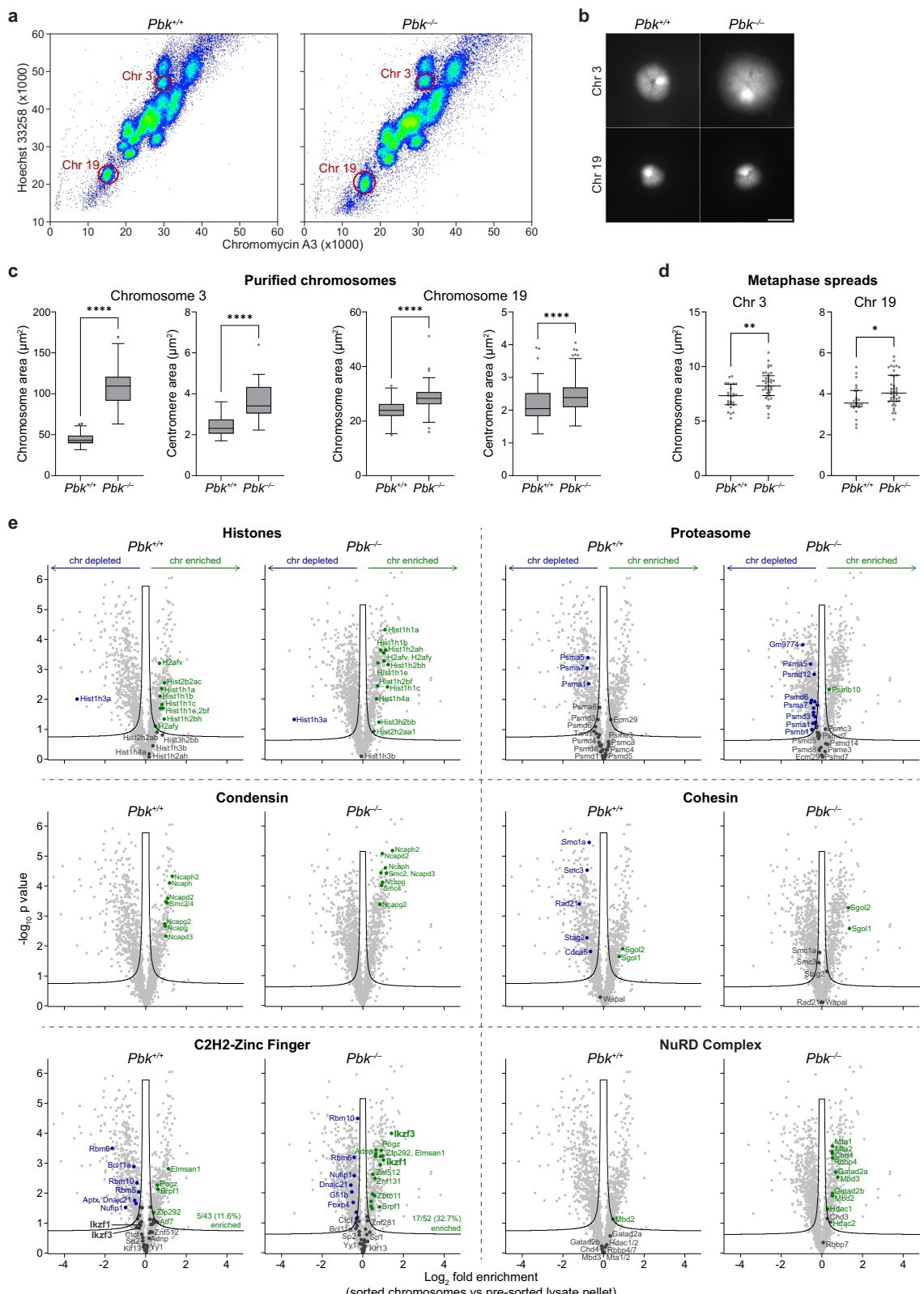

performed motif enrichment analysis at differentially accessible sites. Peaks with increased accessibility in *Pbk⁻/⁻* samples were highly enriched for CTCF and other C2H2-ZF motifs (Fig. 5e). In contrast, peaks with decreased accessibility were generally not enriched for C2H2-ZF motifs (Supplementary Fig. 5f); although interestingly, the only C2H2-ZF representative amongst the top ten enriched motifs at these sites corresponded to the repressor ZBTB11, which was found to require

PBK phosphorylation for mitotic eviction (Fig. 4c and Supplementary Fig. 4d).

## Increased mitotic retention of CTCF and altered chromatin in the absence of PBK

In order to probe for evidence of differential DNA binding of individual factors between *Pbk⁺/⁺* and *Pbk⁻/⁻* samples, we performed TF

**Fig. 3 | Loss of PBK results in reduced mitotic chromosome compaction and increased retention of specific factors. a** Mitotic chromosomes from demecolcine-arrested *Pbk*^+/+^ and *Pbk*^−/−^ mouse preB cells visualised by flow cytometry after staining with Hoechst 33258 and Chromomycin A3. Gates for the purification of chromosomes 3 and 19 are shown, representative of three replicates. **b** Flow-purified mitotic chromosomes 3 (upper) and 19 (lower) from arrested *Pbk*^+/+^ and *Pbk*^−/−^ mouse preB cells, cytocentrifuged onto slides and stained with DAPI. Representative of chromosomes from three independent replicates; scale bar = 5 μm. **c** Size quantification of mitotic chromosomes 3 and 19, flow-purified from arrested *Pbk*^+/+^ and *Pbk*^−/−^ mouse preB cells, measuring total (left) and DAPI-dense centromeric (right) areas. Quantification from one experiment, representative of three independent replicates (chromosome 3: *n* = 117, 65 (total area) and *n* = 96, 49 (centromeres); chromosome 19: *n* = 133, 189 (total area) and *n* = 113, 164 (centromeres); boxplots show median, interquartile range and Tukey whiskers;

****$p$ < 0.0001, unpaired two-tailed *t* tests). **d** Size quantification of chromosomes 3 and 19 in metaphase spreads from arrested *Pbk*^+/+^ and *Pbk*^−/−^ mouse preB cells, identified by chromosome paints. Quantification from one experiment, representative of two replicates (*n* = 26, 39 (chr 3) and *n* = 24, 39 (chr 19); plots show median and interquartile range; **$p$ = 0.0017, *$p$ = 0.0172, unpaired two-tailed *t* tests. **e** Volcano plots highlighting specific factors enriched on or depleted from *Pbk*^+/+^ and *Pbk*^−/−^ mitotic chromosomes, compared to pre-sorted lysate pellets (green = enriched, blue = depleted, dark grey = not significantly enriched/depleted; modified two-tailed *t* test with permutation-based false discovery rate (FDR) < 0.05, S0 = 0.1; *n* = 4 chromosome samples and *n* = 3 lysate pellet samples; only selected annotation labels are shown for proteasome and C2H2-zinc finger proteins for clarity; volcano plots highlighting all enriched/depleted factors are shown in Supplementary Fig. 3f). Source data for Fig. 3c, d are provided as a Source Data file.

footprinting analysis on our ATAC-seq data (Fig. 5f). These results revealed increased footprint protection for a wide range of C2H2-ZF proteins, consistent with generally increased retention of these factors on *Pbk*^−/−^ mitotic chromosomes. Although significant, most footprint differences were relatively modest; however, in contrast to these subtle differences, CTCF motifs clearly showed enhanced protection in the *Pbk*^−/−^ samples (Fig. 5f and Supplementary Fig. 5g), consistent with this factor remaining chromosome-associated in mitosis (Fig. 4d). To confirm this, we performed CTCF ChIP-qPCR on asynchronous and purified mitotic *Pbk*^+/+^ and *Pbk*^−/−^ cells (Supplementary Fig. 5h, i). Mitotic cells were FACS-purified based on H3S10 phosphorylation, as illustrated in Supplementary Fig. 5h, enabling CTCF binding to be examined at nine selected loci (six positive and three negative regions, including CTCF peaks shown in Fig. 5d). CTCF was selectively (and similarly) enriched at known binding sites[61] in both *Pbk*^+/+^ and *Pbk*^−/−^ asynchronous cells (Supplementary Fig. 5i, upper). In contrast, CTCF binding at these sites was significantly higher in purified mitotic *Pbk*^−/−^ cells compared to *Pbk*^+/+^ controls, where we detected minimal mitotic retention of CTCF (Supplementary Fig. 5i, lower). Analysing published interphase CTCF-binding sites[61] (Fig. 5g, left), we observed genome-wide increased chromatin accessibility at these loci in *Pbk*^−/−^ compared to *Pbk*^+/+^ mitotic chromosomes (Fig. 5g), in agreement with the motif enrichment results shown in Fig. 5e. Nucleosome positioning around known CTCF-binding sites was also altered on *Pbk*^−/−^ mitotic chromosomes (Fig. 5h), such that flanking nucleosomes were shifted slightly outwards from CTCF motifs (421 bp spacing) as compared with mitotic *Pbk*^+/+^ spacing (399 bp), and more closely resembled the spacing seen in WT asynchronous samples (441 bp). Together, these data are consistent with widespread mitotic dissociation of CTCF in preB cells mediated by PBK phosphorylation, that is effectively hampered in the absence of PBK. Our observations are also in agreement with a previously suggested model whereby mitotic dissociation of CTCF leads to an inward shift of flanking nucleosome positions[62]. Whilst we do not rule out the contribution of other factors, collectively these results suggest that in the absence of PBK, CTCF remains chromosome-associated throughout mitosis and results in a greater preservation of nucleosome-free accessible regions by preventing adjacent nucleosomal array repositioning (Fig. 5i).

## Discussion

It has been proposed that transcription factors that remain associated with chromosomes throughout mitosis could bookmark the genome for future rapid expression of genes in daughter cells as they enter G1[1,13,63–67]. Mitotic bookmarking factors have been described in many different cell types, including ESRRB in mouse embryonic stem cells (mESCs)[14,68], GATA1 in erythroblasts[65], GATA2 in haemopoietic precursors[5], FOXA1 in hepatocytes[66] and RUNX2 in osteoblasts[69]. However, we have previously estimated that only approximately ten percent of proteins that were detected in mitotic mESC lysate pellets were enriched on chromosomes at metaphase[10]. Most chromatin-

bound factors were presumed to dissociate as a consequence of reduced transcription, altered DNA binding kinetics, physical compaction of chromosomes, or in response to mitotic kinase-mediated phosphorylation and eviction[70]. In addition, alterations to the chromatin environment, for example, depletion of H3K9me3, can also alter the retention of specific bookmarking factors on metaphase chromosomes[53]. Here, we investigated the mitotic eviction of the lymphoid-specific DNA-binding factor Ikaros. We showed that a mitotic kinase, PBK (also known as TOPK), that was independently discovered by two groups more than twenty years ago[58,71] and causes global phosphorylation of C2H2-ZF proteins[49,50], was both necessary and sufficient to release Ikaros from mitotic chromosomes. Genetic deletion of PBK from mouse preB cells prevented Ikaros displacement during metaphase. Furthermore, in WT cells, Ikaros rapidly re-associated to metaphase chromosomes when treated in situ with the PBK inhibitor OTS514. This is reminiscent of the rapid re-localisation to centromeric regions we observed at anaphase/telophase in normal cells during mitosis, coincident with the previously reported loss of active PBK in anaphase[50]. Our results, therefore, show that Ikaros recruitment to chromosomes is regulated by a dynamic interplay between the mitotic-specific activity of PBK and an opposing phosphatase, most likely PP1, which colocalises with Ikaros at pericentromeric heterochromatin and regulates binding in interphase[55,72,73]. Furthermore, our data from T cells and macrophages suggest that this mechanism is shared between different cell types.

While PBK can mediate the phosphorylation of many C2H2-ZF proteins, including SP1 and YY1 (as well as many others exemplified in Fig. 4c), it is worth noting that SP1 and YY1 did not show re-localisation to mitotic chromosomes in PBK-null cells (despite loss of linker phosphorylation). Although ATAC-seq footprinting analysis detected differences at multiple C2H2-ZF motifs, most changes were subtle, suggesting relatively minor alterations in the occupancy or affinity of these factors in the absence of PBK. This implies that many targets of PBK are also regulated by other kinases, which can phosphorylate these factors at additional sites and displace them from mitotic chromosomes. This hypothesis is consistent with published literature; for example, YY1 is reported to be targeted by Aurora kinases[18,19], whilst DNA-binding of SP1 in mitosis has been shown to be regulated by CDK1[20,21] (both at non-linker sites). Interestingly, we also noted that SP1 and YY1 share a highly similar potential Aurora A target motif. In contrast to this, a subset of C2H2-ZF factors, including Ikaros, Aiolos, CTCF, ZNF131 and ZBTB11, which all lack putative Aurora A target motifs, appear to be uniquely dependent on PBK for mitotic displacement. Such distinctions may be important when considering the likely impacts of PBK/TOPK inhibitors in future cancer therapeutics[74].

Among the C2H2-ZF targets of PBK-mediated phosphorylation, we identified CTCF. In dividing preB cells, we show that CTCF is largely displaced from chromosomes by metaphase but remains chromosome-associated in PBK-null cells (at its interphase binding sites), demonstrating unequivocally that displacement of CTCF from

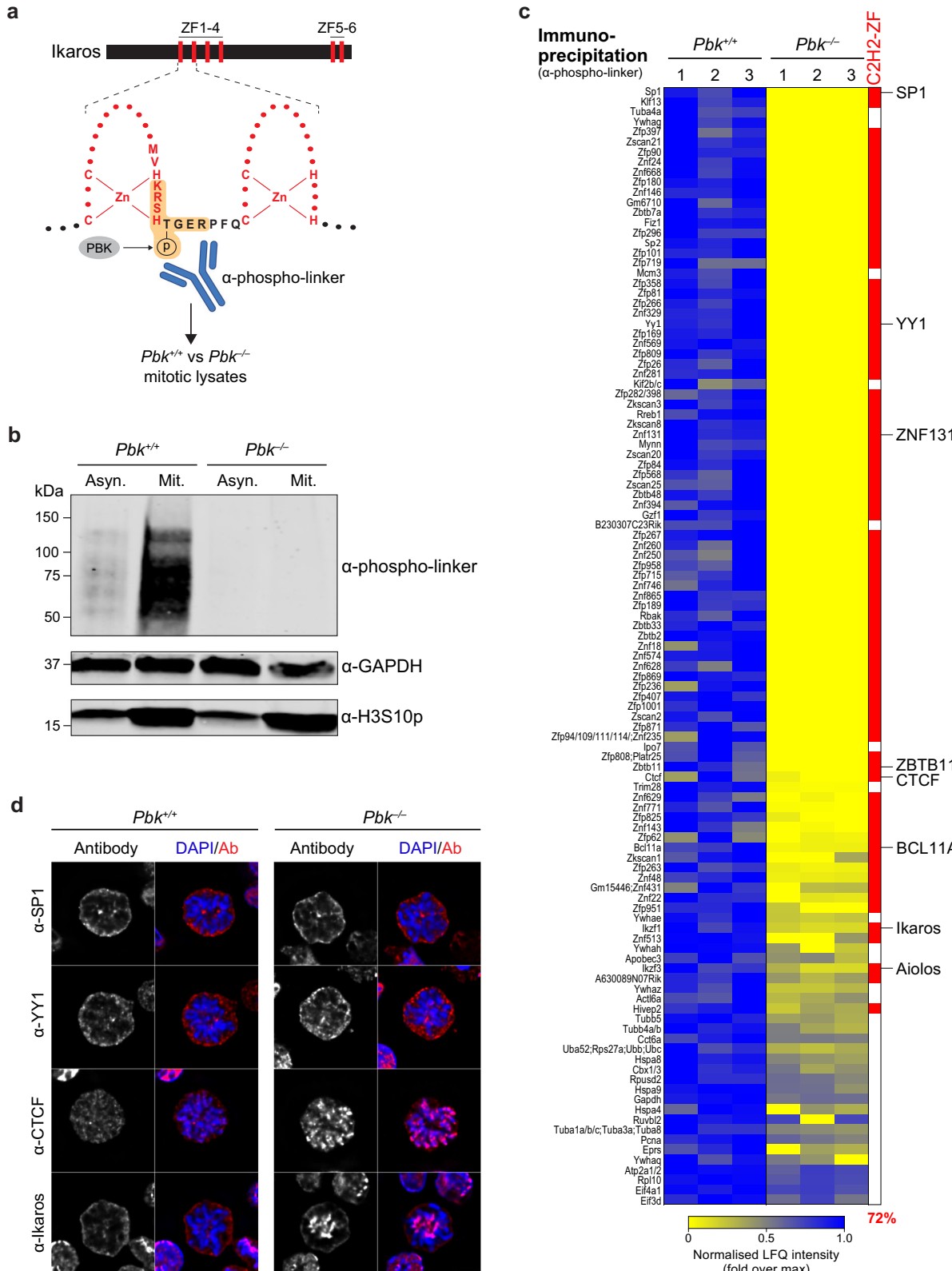

chromosomes is regulated by PBK. Interestingly, CTCF re-associates with slightly slower kinetics than Ikaros during normal mitotic exit, or following acute PBK inhibition, highlighting how dynamic binding through the cell cycle must also depend on a variety of other parameters such as the rate of dephosphorylation, intrinsic binding properties and the chromatin environment of target loci. CTCF can also stabilise the binding of cohesin complexes[75,76], and in this context, it is

interesting to note that mitotic depletion of cohesin appears less pronounced in *Pbk*[−/−] cells where CTCF is retained. Intriguingly, despite more than a decade of intense investigation, the status of CTCF in interphase[77,78] and as a mitotic bookmark[3,12,62,67,75,79–82], remains enigmatic. In interphase, CTCF plays a key role in forming topologically associated domains (TADs) and demarcating chromatin loops[76,78,83–85], but by prometaphase, such structures are no longer detected[62,82,86] and

**Fig. 4 | PBK phosphorylates and regulates multiple C2H2-ZF proteins in mitosis. a** Schematic of the Ikaros protein illustrating positions of zinc fingers (ZFs) 1–4, involved in DNA binding, and ZFs 5–6, involved in dimerisation. ZFs 1 and 2 (red) and the conserved linker sequence (black) are shown in more detail, along with the proposed PBK mitotic phosphorylation site[16]. An anti-phospho-linker antibody was generated against a phosphorylated peptide corresponding to the shaded sequence and used to probe mitotic lysates from $Pbk^{+/+}$ or $Pbk^{-/-}$ mouse preB cells. **b** Western blot detection of proteins recognised by anti-phospho-linker antibody in asynchronous (Asyn.) and mitotically arrested (Mit.) $Pbk^{+/+}$ and $Pbk^{-/-}$ mouse preB cell lysates. GAPDH was used as a loading control, and H3S10p was included as a mitotic marker. Representative of three independent experiments, with results further validated in a second PBK KO clone. Source data are provided as a Source Data file. **c** Heatmap representation of LC-MS/MS analysis of proteins immunoprecipitated with anti-phospho-linker antibody from $Pbk^{+/+}$ mitotic lysates, compared to $Pbk^{-/-}$ control mitotic lysates ($n = 3 + 3$). Normalised LFQ intensity values,

for three independent replicates, are shown for factors (111) which were detected after IP from $Pbk^{+/+}$ mitotic lysates, but which were undetected or at significantly lower levels after IP from $Pbk^{-/-}$ lysates (modified two-tailed $t$ test with permutation-based FDR < 0.05 and S0 = 0.1; see Methods for full filtering criteria). Proteins containing a C2H2-ZF domain are highlighted in red and constitute 72% of identified candidates. **d** Immunofluorescence staining of the indicated factors in fixed WT ($Pbk^{+/+}$) or $Pbk^{-/-}$ mitotic mouse preB cells from asynchronously dividing cultures. Images are representative of > 16 mitotic cells from across two (YY1, CTCF) or three (SP1, Ikaros) independent staining experiments. Staining patterns were further validated in a second PBK KO clone (SP1, CTCF, Ikaros) and with an alternative antibody (SP1). Ikaros staining is derived from the experiment shown in Fig. 2d and a further representative image is included here for comparison. Scale bar = 5 μm. **a** Partially created in BioRender. Dimond, A. (2025) https://BioRender.com/x87bt2h.

CTCF is generally thought to largely be dissociated[7,62,87,88], although the extent of dissociation may be cell-type dependent[89]. Several reports have suggested that CTCF can bookmark specific sites through mitosis, at least in certain cells[3,67,79,80,89], and it is proposed that this could render candidate genes more susceptible to rapid reactivation in G1[3,67]. Our data are not necessarily at odds with such claims, but do indicate that the majority of CTCF is efficiently evicted from WT mouse preB cell chromosomes by metaphase. ATAC-seq comparisons between $Pbk^{+/+}$ and $Pbk^{-/-}$ mitotic chromosomes revealed enhanced footprint protection, increased accessibility and altered nucleosome positioning at CTCF binding sites in the absence of PBK, consistent with mitotic retention of CTCF at these sites in $Pbk^{-/-}$ cells, although additional factors may contribute to the changes in accessibility. Interestingly, prior reports in which a degron was used to remove CTCF during mitosis did not show widespread reduction in the rapid re-expression of proposed CTCF bookmarked genes[67]. It has subsequently been proposed that mitotic bookmarking of the genome could be achieved by conjoint factors that effectively recognise similar or identical binding sites and operate (in tandem) throughout the cell cycle[6]. In this regard, we have previously shown that ADNP and ADNP2 are retained by mitotic mESC chromosomes[10] (illustrated in Supplementary Fig. 6 for reference), and these factors form part of the ChAHP complex which can compete with CTCF for a common set of binding motifs[90,91]. Whether there is potential overlap in mitotic binding, and therefore potential redundancy of CTCF and ADNP functions in the cell cycle, will be of interest to explore in the future.

Mitotic chromosomes isolated from preB cells that lacked PBK were significantly larger (less compact) than their normal counterparts. This was demonstrated both with purified individual chromosomes 3 and 19 that were stained with Hoechst and Chromomycin dyes and isolated by flow cytometry, or by direct analysis of these chromosomes on metaphase spreads. Although PBK has previously been suggested to contribute to chromosome condensation via H3S10 phosphorylation[60], we did not observe any reduction in this modification in $Pbk^{-/-}$ mitotically arrested cells. These observations, together with widespread increases chromatin accessibility, imply that a failure to correctly displace C2H2-ZF binding proteins at prophase/metaphase may contribute to compromised mitotic chromosome compaction and structure, a result which is broadly in line with claims that chromosome condensation in mitosis requires global reductions in transcription, acetylation and transcription factor binding[70,92]. In addition to C2H2-ZF factors, we also observed an apparent increased retention of other chromatin-related proteins, in particular nucleosome remodelling complexes, an observation which could potentially be explained by TF interactions, such as between Ikaros and NuRD[41–44]. The observation that $Pbk^{-/-}$ chromosomes appear to retain more heterochromatin proteins and constituents of the NuRD deacetylase complex, yet are less compact, may seem counterintuitive. Previously, we have shown that several mediators of repressive chromatin, such as DNA methylation

and PRC2-mediated histone H3K27me3, remained chromosome-associated throughout mitosis, and their deletion resulted in mitotic chromosomes being less condensed than their normal counterparts[10]. On the other hand, removal of the heterochromatin mark H3K9me3 resulted in mitotic chromosomes which were considerably smaller and more compact than their WT equivalents[53]. In addition, cohesin, a complex which is largely removed from chromosome arms during prophase but is protected at centromeres by shugoshin[59], has been shown to provoke a widespread de-condensation of metaphase chromosomes when experimentally cleaved in situ[10]. Collectively, these data caution against overly simplistic predictions of the direct relationship between individual chromatin components and mitotic chromosome condensation and argue that even low levels of chromosome-associated proteins can have important functional and structural consequences for metaphase chromosomes. Despite our observations of a chromosome compaction defect, PBK-deficient cells were able to divide, and asynchronous cultures display a similar cell-cycle distribution to WT cells. Whilst perhaps surprising, this is not dissimilar to other mutants that affect chromosome compaction, which are able to divide in vitro[10,53], but exhibit profound defects in vivo[93]. Early indications are that $Pbk^{-/-}$ cells divide slightly more slowly, and we observed evidence of increased apoptosis during mitotic arrest (as indicated by a small subG1 peak seen in Supplementary Fig. 4b). Further work will be required to characterise the biological impacts of PBK-dependent alterations in chromosome compaction and transcription factor retention.

Ikaros/Lyf-1 is one of the earliest and best studied regulators of lymphoid identity, that was initially discovered and characterised more than thirty years ago[25–27]. Since then, it has become clear that Ikaros regulates lineage-specific gene activity and silencing across multiple scales and levels of organisation, harnessing partnerships with a plethora of other transcription factors, family members and chromatin-modifier complexes[31–46,94]. Ikaros is essential for the commitment and differentiation of lymphocytes[28,30], and its deregulation can drive the onset and progression of B and T cell leukaemias[95–98]. Here we show that the normal displacement of Ikaros (and its homologue Aiolos) from chromosomes entering mitosis is controlled by a single mitotic kinase, PBK, and demonstrate that this is both necessary and sufficient for eviction. This mechanism is shared between cell types, and PBK is also implicated in regulating the mitotic behaviour of CTCF and the NuRD complex, as well as ZNF131 and ZBTB11, two factors that have been shown to prevent the aberrant induction of pro-differentiation genes in pluripotent stem cells[99]. While it is perhaps surprising that mitotic displacement of CTCF and lineage-specific transcription factors from chromosomes relies solely on a single kinase, this intimate dependency could offer the opportunity for coordinated displacement and dynamic rebinding of a vital subset of PBK targets that will ultimately determine the genomic organisation, acetylation, spatial context and expression of lineage-specifying genes early in G1-phase.

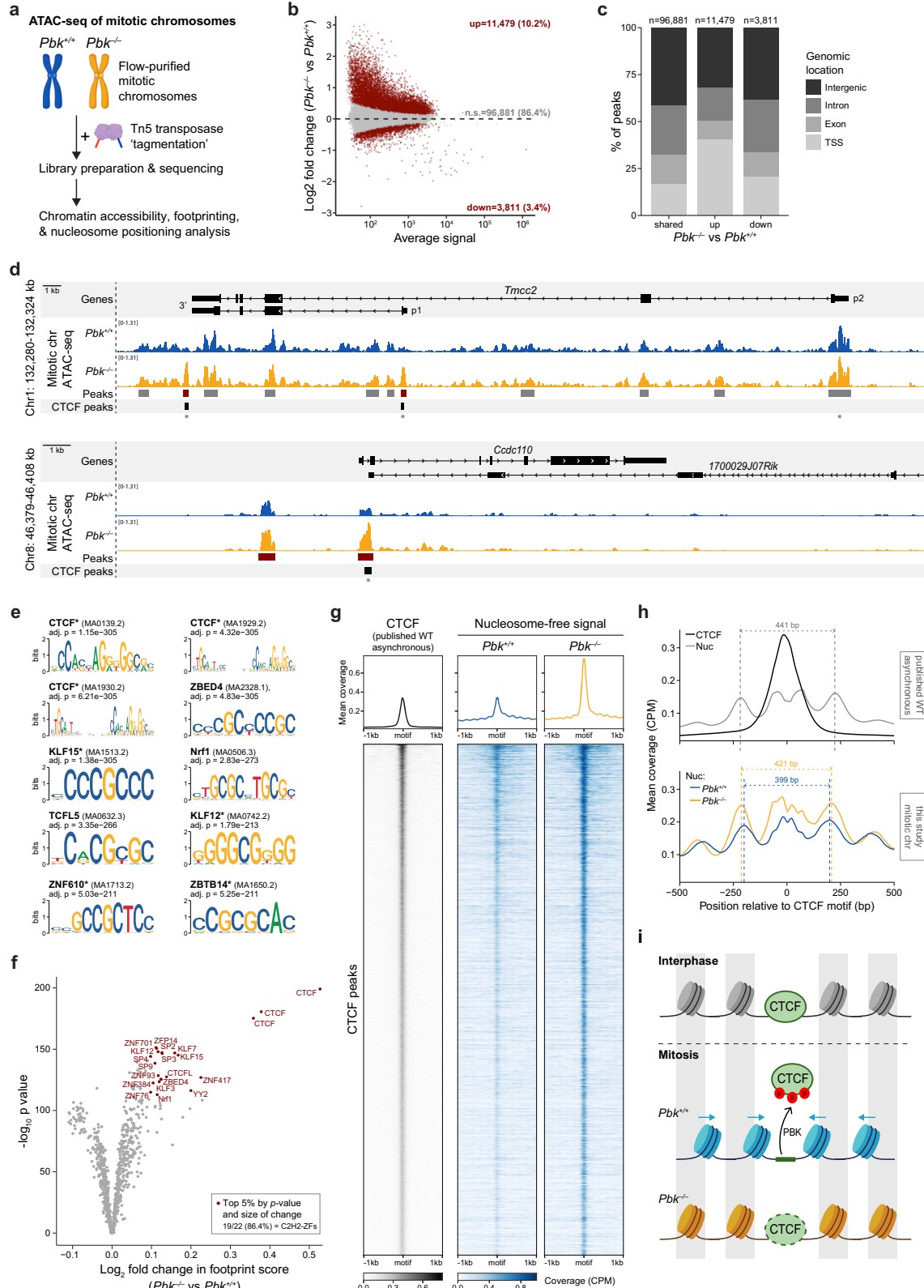

## Methods

### Cells and cell culture

Abelson-transformed WT mouse preB cells (previously generated and used in our lab[10,100]) and genetically modified derivatives (see below) were cultured in suspension in IMDM medium supplemented with 12% FCS, 2 mM L-glutamine, 2X non-essential amino acids, antibiotics and 50 μM 2-mercaptoethanol. Mouse VL3-3M2 cells (gifted by Stephen Smale) were cultured in suspension in IMDM medium supplemented with 10% FCS, 2 mM L-glutamine, 1 mM sodium pyruvate, antibiotics and 50 μM 2-mercaptoethanol. Adherent mouse J774A.1 cells (ATCC, TIB-67) were cultured in DMEM medium supplemented with 10% FCS, 1 mM sodium pyruvate and antibiotics, and were detached with trypsin for passaging. All cells were maintained at 37 °C with 5% $CO_2$ and split every 2–3 days.

**Fig. 5 | Mitotic chromosomes from _Pbk⁻/⁻_ cells have higher chromatin accessibility and show evidence of increased CTCF retention. a** Approach to compare flow-purified mitotic chromosomes from arrested _Pbk⁺/⁺_ and _Pbk⁻/⁻_ mouse preB cells by ATAC-seq. **b** MA plot highlighting differentially accessible ATAC-seq peaks in _Pbk⁻/⁻_ versus _Pbk⁺/⁺_ mitotic chromosomes ($n = 4 + 4$; DESeq2: two-sided Wald test, Benjamini-Hochberg correction, $p_{adj} < 0.1$, n.s. = not significant). **c** Genomic distribution of unchanged or significantly altered ATAC-seq peaks from Fig. 5b. **d** Representative loci showing nucleosome-free ($\leq 100$ bp) ATAC-seq signal from _Pbk⁺/⁺_ (blue) and _Pbk⁻/⁻_ (orange) mitotic chromosomes (normalised merged signal, $n = 4 + 4$). Consensus ATAC-seq peaks (red = differentially accessible) and published CTCF peaks (asynchronous mouse preB cells[61]) shown underneath; asterisks mark loci selected for CTCF ChIP-qPCR (Supplementary Fig. 5i). **e** Top ten enriched motifs in ATAC-seq peaks with increased accessibility in _Pbk⁻/⁻_ mitotic chromosomes; asterisks indicate C2H2-ZF proteins. **f** Differential footprinting analysis (TOBIAS) of _Pbk⁻/⁻_ versus _Pbk⁺/⁺_ mitotic chromosome ATAC-seq data. Motifs in the top 5% by both size of change and _p_-value are highlighted with TF labels (BINDetect = two-sided permutation test); all except ZBED4, Nrf1 and Thap11 (label omitted for clarity) are C2H2-ZF proteins. **g** Nucleosome-free ($\leq 100$ bp) ATAC-seq signal from _Pbk⁺/⁺_ and _Pbk⁻/⁻_ mitotic chromosomes at known CTCF peaks, centred on CTCF motifs. CTCF peaks and ChIP-seq coverage are from asynchronous mouse preB cells[61]. Top = average signal, below = individual peak coverage ($n = 8532$ peaks, ordered by total CTCF signal). **h** Average nucleosomal (180–250 bp) ATAC-seq coverage at CTCF peaks, centred on CTCF motifs, for asynchronous WT mouse preB cells[61] and _Pbk⁺/⁺_ and _Pbk⁻/⁻_ mitotic chromosomes. CTCF ChIP-seq coverage shown for WT asynchronous cells; dotted lines indicate average mid-points of flanking nucleosomes (separation: WT asynchronous = 441 bp, _Pbk⁺/⁺_ mitotic chromosomes = 399 bp, _Pbk⁻/⁻_ mitotic chromosomes = 421 bp). **i** Model for how PBK regulates CTCF dissociation, chromatin accessibility and nucleosome repositioning in mitosis. CTCF binds in interphase, maintaining an accessible nucleosome-free region (NFR), with well-positioned flanking nucleosomes. In WT mitotic cells, PBK phosphorylates and evicts CTCF, leading to nucleosomes shifting inwards and reduced accessibility, consistent with previous work[3,62]. In _Pbk⁻/⁻_ mitotic cells, CTCF remains unphosphorylated and DNA-bound, resulting in NFR protection and reduced repositioning of nucleosomes. **a, i** Partially created in BioRender. Dimond, A. (2025) https://BioRender.com/c5jzd9a, https://BioRender.com/hk53685.

## Mitotic arrest and propidium iodide (PI) staining
PreB cells were diluted to approximately $10^6$ cells/ml and arrested by the addition of 0.1 µg/ml demecolcine (Sigma-Aldrich, D1925) for 5 h. For propidium iodide (PI) staining, $10^6$ cells were fixed with ice-cold 70% ethanol and stored at negative 20 °C until staining. Fixed cells were washed once with PBS and incubated with PI stain (1X PBS, 0.05 mg/ml PI (Sigma-Aldrich, P4864), 1 mg/ml RNase A, 0.05% IGEPAL CA-630) for 10 min at room temperature (RT) and 20 min on ice. PI signal was acquired in linear mode using a BD FACSymphony A3 flow cytometer and BD FACSDiva Software (v9.1). FlowJo software (v10.8.1) was used to analyse the data and create plots.

## CRISPR/Cas9 generation of knock-in (KI) and knock-out (KO) mouse preB cells
The UCSC genome browser (https://genome.ucsc.edu/) was used to select guide RNA sequences (Supplementary Table 1)[101,102]. Corresponding oligos were annealed, phosphorylated with T4 Polynucleotide Kinase (NEB, M0201S) and cloned into the pU6-(BbsI)_CBh-Cas9-T2A-mCherry plasmid[103] (a gift from Ralf Kuehn, Addgene plasmid #64324) by golden gate assembly using BbsI-HF (NEB, R3539S) and T4 DNA ligase (NEB, M0202S). A donor plasmid for _Ikzf1-mNeonGreen_ KI was generated by incorporating the _mNeonGreen_ sequence, flanked by left and right _Ikzf1_ homology arms, into a backbone containing EBFP2 (details in Supplementary Table 1). The NEBuilder Assembly Tool (v2.3.0) was used to design primers for amplification of each component of the donor plasmid (Supplementary Table 1) using Phusion (NEB, M0530S); purified PCR products were assembled using NEBuilder HiFi DNA Assembly Master Mix (NEB, E5520S). Assembled sgRNA/Cas9 and donor plasmids were transformed into DH5α bacteria, and the final purified plasmids were checked by Sanger sequencing (Genewiz).

PreB cells were transfected with the appropriate sgRNA/Cas9 plasmid alone (KO), or in combination with the _mNeonGreen_ donor plasmid (KI), using Lipofectamine 3000 (Invitrogen, L3000008). After 24 h, mCherry-positive or mCherry/EBFP double-positive transfected cells were isolated by Fluorescence-Activated Cell Sorting (FACS, note that efficiency was very low) and cultured as a pool for 1–2 weeks before sorting single cells into 96-well plates. For _mNeonGreen_ KI, cells which were positive for mNeonGreen and negative for both mCherry and EBFP were selected. For _Pbk_ KO, mCherry-negative cells were selected. Clones were grown and screened by PCR (primers in Supplementary Table 1) and/or western blotting. Genetic modifications were verified by Sanger sequencing of PCR products (Genewiz), and clones selected for downstream experiments were confirmed to be karyotypically normal.

## Immunofluorescence (IF)
PreB and VL3-3M2 suspension cells were spun onto poly-D-lysine-coated coverslips or µ-Slide 18-well chamber slides (ibidi), whilst J774A.1 adherent cells were grown directly on µ-Slide 18-well chamber slides. For inhibitor treatments, inhibitor-containing medium was added to the cells for 10 min at 37 °C immediately prior to fixation. Cells were fixed with 2% formaldehyde for 15 min at RT, permeabilised with 0.1–0.5% Triton X-100 for 15 min at RT and blocked with 2% bovine serum albumin and 5% normal goat serum. Primary antibodies (Supplementary Table 2) were diluted in blocking buffer and incubated with cells overnight at 4 °C. Cells were washed three times with PBS and incubated with goat anti-rabbit secondary antibody (Alexa Fluor 633, Invitrogen, A-21070), diluted 1:500 in blocking solution, for 1 h at RT. Cells were washed with PBS and either stained for 5 min with 1 µg/ml DAPI (for chamber slides) or mounted using DAPI-containing Vectashield (Vector Laboratories, H-1200-10; for coverslips). An Olympus IX70 inverted microscope with a UPlanApo 100X/1.35 oil iris objective was used for image acquisition with Micro-Manager software (v2.0). Z-stacks were collected with 0.35 µm intervals, and images were deconvolved using Huygens Professional software (Scientific Volume Imaging, v19.10), using the CMLE algorithm and default parameters for widefield deconvolution. Representative images were processed in Fiji (1.54e)[104] to prepare display figures.

## Live-cell imaging
Ikaros-mNeonGreen KI mouse preB cells were switched to fully supplemented phenol-free medium containing 20 mM HEPES and 0.5–1 µM SiR-DNA (Spirochrome, Tebu bio, SC007) at least 30 min prior to imaging. Cells were transferred to µ-Slide 8- or 18-well chamber slides (ibidi, 80806 and 81816) and allowed to settle or were briefly centrifuged at $300 \times g$ before imaging. For testing different inhibitor treatments (Supplementary Table 3), inhibitors were added immediately prior to transferring cells to chamber slides, and snapshots were collected in a window between 5 min and 15 min after inhibitor addition (average 10 min). For time-lapse imaging of individual cells before and after OTS514 addition, a cell in metaphase was identified and imaged. The medium was then carefully removed and replaced with OTS514-containing medium, and, after checking the focus, another snapshot was acquired immediately, and at 1 min intervals thereafter. Live-cell snapshots were collected on an Olympus IX70 inverted microscope with a 37 °C environmental chamber and 5% $CO_2$ supply using a UPlanApo 100X/1.35 oil iris objective and Micro-Manager software (v2.0). Z-stacks were collected at 0.35 µm intervals, and images were deconvolved as for IF. Following deconvolution, volume analysis was performed using the surfaces and cells packages in Imaris (Bitplane, v10.0.0), and mean intensity measurements were performed

in both chromosomal and cytoplasmic compartments. Statistics were exported from Imaris, collated using a Python script, and the chromosomal to cytoplasmic intensity ratio was calculated; GraphPad Prism (v10.1.1) was used to prepare graphs and perform statistical comparisons. Representative images were processed in Fiji (1.54e)[104] to prepare display figures.

Time-lapse images for Supplementary Movies 1–4 were collected using an Olympus IX83 microscope equipped with a Yokogawa CSU-W1 spinning disk and Hamamatsu ORCA-Flash 4.0 camera, with a 37 °C environmental chamber and 5% $CO_2$ supply, using a UPlanSApo 60x/1.35 oil objective. Z-stacks (3 μm step size) were acquired with cellSens Dimension software (v2.3) every 3 min, with autofocus enabled and laser power set at 2%. Time-lapse videos were prepared in Fiji (1.54e)[104].

## Western blotting

Asynchronous or arrested preB cells were pelleted at $300\,g$ and either processed immediately or snap-frozen and stored at negative 80 °C. For OTS514 treatment, 10 μM of inhibitor was added to arrested cells for 10 min at 37 °C, before pelleting cells and processing immediately. For processing, cells were resuspended in cold RIPA buffer (50 mM Tris-HCl pH 8.8, 150 mM NaCl, 1% Triton X-100, 0.5% sodium deoxycholate, 0.1% SDS, 1 mM EDTA, 3 mM $MgCl_2$) containing 250 μ/ml Benzonase (Sigma, E1014), 1X cOmplete EDTA-free protease inhibitor cocktail (Roche, 11873580001), and, for detection of phosphorylated proteins, 1X PhosSTOP phosphatase inhibitor (Roche, 4906845001). Samples were incubated at RT for 20 min before quantification with the *DC* protein assay (Bio-Rad, 5000116). Samples were diluted with RIPA buffer, if required, and mixed 3:1 with 4X Laemmli buffer (250 mM Tris-HCl, pH 6.8, 8% SDS, 40% glycerol, 20% β-mercaptoethanol, bromophenol blue). Proteins (15–30 μg) were resolved on a 4–15% or 10% acrylamide gel and transferred onto a PVDF membrane (Invitrogen iBlot 2 or Bio-Rad Trans-Blot Turbo system). Membranes were blocked with 5% milk (standard protein detection) or 5% BSA (phosphorylated protein detection) in TBS-T (TBS with 0.1% Tween 20) for 1 h at RT before incubating with primary antibodies (Supplementary Table 2) overnight in blocking solution. After three washes in TBS-T, membranes were incubated with appropriate secondary antibody (Supplementary Table 2) for 1 h at RT in blocking solution. Following three washes in TBS-T, fluorescence imaging of the membrane was performed with the LI-COR Odyssey CLx or Fc systems using Image Studio software (v5.2.5 or v5.5). Images were processed with Image Studio Lite (v5.2).

## Chromosome size measurements from metaphase spreads

Arrested preB cells were lysed in hypotonic solution (75 mM KCl, 10 mM $MgSO_4$, pH 8) for 25 min at 37 °C, before centrifuging at $500\,g$ for 8 min to pellet nuclei. Nuclei were resuspended in residual supernatant and stored in fixative (75% methanol, 25% acetic acid) at negative 20 °C. To prepare metaphase spreads, nuclei were pelleted ($500\,g$, 8 min), washed three times in fixative, and resuspended in a small volume of fixative (to a pale grey appearance). Next, 23 μl of sample was dropped directly onto a 20 μl drop of 45% acetic acid on a glass Twinfrost slide, tilting to spread nuclei, before air-drying slides at RT.

Chromosomes 3 and 19 were detected with mouse chromosome paints (MetaSystems Probes, D-1403-050-OR and D-1419-050-FI) according to the manufacturer's protocol before samples were stained with DAPI (1 μg/ml, 5 min) and mounted in Vectashield (Vector Laboratories, H-1000-10). Images were collected with a Leica SP5 II confocal microscope using LAS-AF software (v2.7.3.9723). Chromosome spread images were first segmented using Cellpose (v2.0)[105] with the 'cyto' model, with additional training performed to improve the segmentation precision of individual chromosomes in clumped spreads. Labelled mask images were subsequently imported into QuPath (v0.4.3)[106] for chromosome paint classification and area

measurements. GraphPad Prism (v10.1.1) was used to prepare graphs and perform statistical comparisons; representative images were processed in Fiji (1.54e)[104] to prepare display figures.

## Isolation of mitotic chromosomes by flow cytometry

Mitotic chromosomes were purified by flow cytometry as previously described[10,53], with minor modifications. Briefly, arrested preB cells (~ $10^8$) were incubated for 20 min at RT in 10 ml of hypotonic solution (75 mM KCl, 10 mM MgSO4, 0.5 mM spermidine trihydrochloride (Sigma-Aldrich, S2501), 0.2 mM spermine tetrahydrochloride (Sigma-Aldrich, S2876), pH 8.0), followed by 15 min on ice in 3 ml of polyamine buffer (80 mM KCl, 15 mM Tris-HCl, 2 mM EDTA, 0.5 mM EGTA, 3 mM DTT, 0.25% Triton X-100, 0.5 mM spermidine trihydrochloride, 0.2 mM spermine tetrahydrochloride, pH 7.7). Samples were vortexed for 30 s at maximum speed, passed several times through a 21-gauge needle, centrifuged at $200\,g$ for 2 min, filtered through a 20 μm CellTrics filter (Sysmex, 04-004-2325) and stored at 4 °C overnight. The next day, chromosomes were stained on ice by adding 5 μg/ml Hoechst 33258 (Sigma-Aldrich, 94403), 25 μg/ml Chromomycin A3 (Sigma-Aldrich, C2659), and $MgSO_4$ (10 mM final) for 45 min, followed by addition of sodium citrate (10 mM final) and sodium sulphite (25 mM final) for 1 h.

Stained chromosomes were purified as reported[10,53], using a Becton Dickinson Influx with BD FACS Sortware (v1.2.0.142), using a 70 μm nozzle tip, a drop-drive frequency of 96 kHz and a sheath pressure of 448 kPa. Forward scatter was measured with a 488 nm laser (Coherent Sapphire, 200 mW); Hoechst 33258 was analysed with a 355 nm air-cooled laser (Spectra Physics Vanguard, 350 mW) and 400 nm (long-pass)/500 nm (short-pass) filters; Chromomycin A3 was analysed with a water-cooled 460 nm laser (Coherent Genesis, 500 mW) and 500 nm (long-pass)/600 nm (short-pass) filters.

## Size measurements of flow-sorted chromosomes

Purified chromosomes 3 or 19 were cytocentrifuged (Cytospin3, Shandon) at $163\,g$ for 10 min onto Polysine slides (VWR, 631-0107). After mounting with Vectashield containing DAPI (Vector Laboratories, H-1200-10), chromosomes in the centre of the slide were imaged on an Olympus IX70 inverted microscope using a UPlanApo 100X/1.35 oil iris objective and Micro-Manager software (v2.0). Chromosome size analysis was performed in Fiji (1.54e)[104]: following background subtraction and median filtering, images were automatically thresholded and binarised for individual chromosome segmentation and subsequent area measurements. Images were manually inspected to exclude misidentified chromosomes or centromeres. GraphPad Prism (v10.1.1) was used to prepare graphs and perform statistical comparisons; representative images were processed in Fiji (1.54e)[104] to prepare display figures.

## LC-MS/MS analysis of mitotic chromosome samples

Flow-purified chromosomes ($10^7$) or unpurified chromosomes from the pre-sorted lysate (85–100 μl) were pelleted at $10,000\,g$ for 10 min at 4 °C; pellets were snap-frozen and stored at negative 80 °C. Pellets were processed by trypsin digest using the iST Sample Preparation Kit (PreOmics, P.O.00001), with minor modifications to the manufacturer's protocol as follows: in step 1.1, 30 μl lysis buffer was used and samples were sonicated in an ultrasonic bath (1 min) both before and after heating; in steps 3.8–9, samples were resuspended in 20 μl and subjected to both sonication and shaking.

LC-MS/MS analysis was performed using an UltiMate 3000 RSLC nano-flow liquid chromatography system (Thermo Scientific) coupled to a Q-Exactive HF-X mass spectrometer (Thermo Scientific) via an EASY-Spray source (Thermo Scientific) as previously reported[10], with minor variations as follows. Samples were loaded onto a trap column (Acclaim PepMap 100 C18, 100 μm x 150 mm) at 10 μl/min in 2% acetonitrile and 0.1% trifluoroacetic acid, eluted to an analytical column (Acclaim Pepmap 100 C18, 75 μm x 50 cm) at 250 nl/min, and

separated using an increasing gradient of buffer B (75% acetonitrile, 5% DMSO, 0.1% formic acid) in buffer A (5% DMSO, 0.1% formic acid): 1–5% for 5 min, 5–22% for 70 min, 22–42% for 20 min. Eluted peptides were analysed by data-dependent acquisition as described[10], except with maximum injection time for MS2 of 110 ms. For quality control purposes, pooled replicate samples (pre-sorted or purified chromosome replicates were pooled separately) were run at the start, middle and end to verify consistency throughout the run.

Raw data were processed with MaxQuant (v1.6.10.43)[107], using the in-built Andromeda search engine against the UniProt *Mus musculus* one-gene-per-protein database (v20230131; 21,976 entries) and against a universal protein contaminants database[108] (downloaded 20220604; 381 entries). A reverse decoy search approach was used at a 1% false discovery rate (FDR) for both peptide spectrum matches and protein groups (parameters included: max missed cleavages = 3; fixed modifications = cysteine carbamidomethylation; variable modifications = methionine oxidation, protein N-terminal acetylation, asparagine deamidation, cyclisation of glutamine to pyro-glutamate). Label-free quantification (LFQ) was performed (LFQ min ratio count = 2) and 'match between runs' was enabled (match time limit = 0.7 min, alignment time limit = 20 min). Two pre-sorted lysate pellet samples (*Pbk*[+/+] replicate 1 and *Pbk*[-/-] replicate 4) exhibited outlier retention time distributions and were excluded from the analysis.

Subsequent data analyses were carried out in Perseus (v1.6.15.0)[109], comparing purified chromosomes (*n* = 4) to pre-sorted lysate pellet (*n* = 3, due to excluded samples) for each of *Pbk*[+/+] and *Pbk*[-/-] conditions to assess chromosomal enrichment or depletion of factors. Proteins with 'only identified by site' or 'reverse' annotations were removed, and data were log2 transformed. Pearson correlation values (LFQ intensities) were extracted using the multi scatter plot function in Perseus and visualised as a heatmap using the R package corrplot (v0.92). Data were filtered in Perseus to keep proteins with ≥ 3 valid replicate LFQ intensities per experimental group (purified chromosomes or pre-sorted lysate pellet); the volcano plot function was used to perform a *t* test (modified two-tailed *t* test: permutation-based FDR < 0.05, randomisations = 250, S0 = 0.1) and for subsequent visualisations. Subsets of proteins were highlighted on volcano plots according to GO term and PROSITE annotations assigned in Perseus (mainAnnot.mus_musculus.txt), with additional manual curation using the MGI Gene Ontology Browser (https://www.informatics.jax.org/function.shtml) where required. The following definitions were used: Histones=gene name contains 'Hist' or 'H2a' (15 in *Pbk*[+/+]; 14 in *Pbk*[-/-]); Proteasome = GOCC name contains 'proteasome' (32 in *Pbk*[+/+]; 36 in *Pbk*[-/-]); Cohesin = GOCC name 'Cohesin complex' plus Stag1/2 (8 in *Pbk*[+/+]; 7 in *Pbk*[-/-]); Condensin = GOCC name 'Condensin complex' or protein name contains 'condensin' (8 in *Pbk*[+/+]; 8 in *Pbk*[-/-]); C2H2-Zinc Finger = PROSITE PS00028 or PS50157 (43 in *Pbk*[+/+]; 52 in *Pbk*[-/-]); NuRD Complex = GOCC name 'NuRD complex' plus Mbd2 and GATAD2b (11 in *Pbk*[+/+]; 12 in *Pbk*[-/-]).

GO term overrepresentation analysis was performed using PANTHER[110] (https://www.pantherdb.org/; v18.0; Fisher's exact test with FDR correction; GO Ontology database https://doi.org/10.5281/zenodo.8436609; GO cellular component complete). Analysis was performed for factors enriched or depleted on mitotic chromosomes (run together), for each of *Pbk*[+/+] and *Pbk*[-/-] conditions (run separately), using the total proteins detected in each condition as background (all proteins displayed on respective volcano plots). The first majority protein ID per hit was used as input to PANTHER, with unassigned IDs manually updated using the second ID or gene name where possible (*Pbk*[+/+] mapped IDs: background = 2007, enriched = 341, depleted = 503; *Pbk*[-/-] mapped IDs: background = 2281, enriched = 400, depleted = 679). GO terms were filtered for significance (FDR < 0.05) in at least one condition, and the top-most overrepresented terms in either condition were selected for visualisation (14 terms each for mitotically enriched or depleted factors), ordered by the difference in fold overrepresentation between *Pbk*[+/+] and *Pbk*[-/-] conditions.

## Immunoprecipitation (IP)

Arrested preB cells (~ 3 × 10^7) were washed twice in ice-cold PBS, snap-frozen and stored at negative 80 °C until processing. Cells were lysed at RT for 10 min in 1 ml RIPA buffer containing 250 μ/ml Benzonase (Sigma), 1X cOmplete EDTA-free protease inhibitor cocktail (Roche, 11873580001), and 1X PhosSTOP phosphatase inhibitor (Roche, 4906845001) before centrifuging at 16,000 × *g* for 10 min at 4 °C. Mitotic lysates were quantified with the Bio-Rad DC protein assay kit and diluted to 0.8 mg/ml with RIPA buffer containing protease and phosphatase inhibitors.

For IP, 1 ml (0.8 mg) of diluted mitotic lysate (input) was incubated overnight with 3.2 μg anti-phospho-linker antibody (Supplementary Table 2) at 4 °C with end-to-end rotation. MagReSyn Protein G beads (ReSyn Biosciences, MR-PRG002) were washed three times in RIPA buffer, and 10 μl of resuspended beads were added to each IP sample and incubated for 10 min at RT, followed by 1 h at 4 °C, with rotation, to capture antibody-antigen complexes. Using a magnetic separator, the supernatant was removed, and the beads were washed five times with RIPA buffer (first wash with protease/phosphatase inhibitors) and three times with 20 mM EPPS, each wash for 5 min at 4 °C, with rotation. In the final wash, samples were transferred to 0.2 ml PCR tubes, all supernatant was removed, and beads were snap-frozen and stored at negative 80 °C until processing for mass spectrometry analysis.

## LC-MS/MS analysis of IP and input samples

**IP sample preparation.** Dry beads underwent on-bead digestion in 50 μl digestion solution (20 ng/μl trypsin (Pierce™, 90059), 10 ng/μl lysyl (WAKO, 129-02541), 50 mM EPPS) for 2 h at 37 °C, followed by removal of solution from the beads and overnight digestion at RT. The next day, desalting was performed using 100 g/l Oasis HLB beads (Waters, 186007549; 100 μg beads to 1 μg peptides) distributed in an OF1100 orochem filter plate and a vacuum manifold. Beads were washed twice with 100 μl acetonitrile and three times with 100 μl 0.1% trifluoroacetic acid (TFA). Samples were acidified to 0.5% TFA, loaded onto a plate, washed twice with 400 μl 0.1% TFA, once with 200 μl $H_2O$, centrifuged at 1000 *g* and eluted in 20 μl 70% acetonitrile. Samples were dried by vacuum centrifugation and resuspended in 30 μl 0.1% TFA for mass spectrometry analysis.

**Input (mitotic lysate) sample preparation.** Samples were prepared for Protein Aggregation Capture[111,112] by mixing samples with SDS (1% final concentration), EPPS pH 8.5 (50 mM final concentration), chloroacetamide (20 mM final concentration) and TCEP (tris(2-carboxyethyl)phosphine, 10 mM final concentration). After heating at 60 °C for 5 min and leaving on the bench for 30 min, prewashed hydroxyl beads (ReSyn Biosciences, MR-HYX010) were added at a protein-to-bead ratio of 1:5. Ethanol was added to a final concentration of 70%, pipetted up and down, left for 5 min to observe bead aggregation, and washed three times with 100 μl 80% ethanol. For digestion, 10 ng/μl trypsin and 5 ng/μl LysC in 100 mM ammonium bicarbonate, pH 8, were added, and the mixture was incubated for 18 h at 37 °C with shaking at 1600 rpm.

**LC-MS/MS analysis.** Chromatographic separation was performed using an UltiMate 3000 RSLC nano-flow liquid chromatography system (Thermo Scientific) coupled to a Q-Exactive mass spectrometer (Thermo Scientific) via an EASY-Spray source (Thermo Scientific). Electro-spray nebulisation was achieved by interfacing to Bruker Pep-Sep emitters (10 μm, PSFSELJ10). Peptide solutions were injected directly onto the analytical column (self-packed column, CSH C18 1.7 μm beads, 300 μm x 30 cm) at a working flow rate of 5 μl/min for 4 min. Peptides were then separated using a 120 min stepped gradient: 0–45% of buffer B (75% MeCN, 5% DMSO, 0.1% formic acid) in buffer A (5% DMSO, 0.1% formic acid), followed by column conditioning and equilibration. Eluted peptides were analysed by the mass spectrometer

in positive polarity, using a data-dependent acquisition mode. An initial MS1 scan (resolution = 140,000; AGC target = 3e6; maximum injection time = 50 ms; range = 400–1800 m/z) was followed by eight MS2 scans (analytes with +1 and unassigned charge state excluded; resolution = 35,000; AGC target = 1e5, maximum injection time = 128 ms; intensity threshold = 2e3). Normalised collision energy was set to 27%, dynamic exclusion was set to 45 s, and total acquisition time was 150 min.

**Data processing.** Raw data were processed with MaxQuant (v1.6.10.43)[107], searching against the UniProt *Mus musculus* one-gene-per-protein database (v20220720; 21,992 entries), using the same parameters as for chromosome LC-MS/MS samples (see above). Label-free quantification was performed (LFQ min ratio count = 1) and 'match between runs' was enabled (match time limit = 0.7 min, alignment time limit = 20 min).

Subsequent data analyses were carried out in Perseus (v1.6.15.0)[109], comparing $Pbk^{+/+}$ versus $Pbk^{-/-}$ input or $Pbk^{+/+}$ versus $Pbk^{-/-}$ IP. Proteins with 'only identified by site', 'reverse' or 'potential contaminants' annotations were removed. Proteins were considered robustly detected in only a single condition ($Pbk^{+/+}$ or $Pbk^{-/-}$) if LFQ > 0 for all three replicates of one condition and LFQ > 0 for a maximum of one replicate of the other condition with a value ≥ 2-fold below the mean of the first condition. Proteins detected in at least five samples (across both conditions) were considered significantly different on the basis of a *t* test (modified two-tailed *t* test: permutation-based FDR < 0.05, randomisations = 250, S0 = 0.1). Candidate PBK targets were defined as those hits which were robustly detected after IP from $Pbk^{+/+}$ but not $Pbk^{-/-}$ lysates, or which were significantly lower after IP from $Pbk^{-/-}$ lysates. There was no overlap between these candidates and those proteins which were significantly higher or only detected in $Pbk^{+/+}$ compared to $Pbk^{-/-}$ input samples. Normalised LFQ values (fold over max) for candidate PBK targets were displayed as a heatmap using conditional formatting in Microsoft Excel. PROSITE annotations were assigned in Perseus (mainAnnot.mus_musculus.txt) by UniProt majority protein IDs, with missing annotations supplemented manually from UniProt (https://www.uniprot.org/). C2H2-ZF proteins were defined by matches to PROSITE accession numbers PS00028 and/or PS50157.

## ATAC-seq libraries

ATAC-seq was performed on mitotic chromosomes[10,53], using chromosomes purified from the same preparations as the proteomics analysis (n = 4). Briefly, $2 \times 10^6$ flow-sorted chromosomes were pelleted at 10,000 *g* for 10 min at 4 °C before tagmentation in a 50 μl reaction (2.5 μl Tagmentase (loaded Tn5 transposase, Diagenode, C01070012), 25 μl 2X Tagmentation buffer (Diagenode, C01019043), 22.5 μl $H_2O$) at 37 °C for 30 min with 1000 rpm shaking. DNA was purified (Qiagen MinElute PCR purification kit, 28004) and PCR amplified (7 cycles; NEBNext High Fidelity master mix, M0541S; primer sequences in Supplementary Table 4). Amplified libraries were purified using 1.8X volume Ampure XP beads (Beckman Coulter, A63880), firstly using 0.5X beads to remove large fragments, before using the remaining volume of beads to capture and purify libraries. Purified ATAC-seq libraries were assessed by Bioanalyzer and quantified by Qubit and KAPA Library Quantification (Roche, 07960140001). Paired-end sequencing (60 bp) was carried out on an Illumina NextSeq 2000 (NextSeq 1000/2000 Control Software v1.4.1.39716; primary analysis RTA v3.9.25; secondary analysis DRAGEN Generate FastQ v3.7.4; reads demultiplexed with bcl2fastq2 v2.20 (allowing 0 mismatches)).

## Sequencing data analysis

**ATAC-seq data processing.** ATAC-seq reads from sorted chromosomes were trimmed with fastp (v0.23.3)[113] to remove adaptors and nucleotides with quality < 20. Reads were aligned to GRCm39 with bwa-mem (v0.7.17)[114] and filtered using samtools (v1.17)[115] to retain only properly paired alignments with a quality score of ≥ 3. PCR duplicates were marked using sambamba (v1.0.1)[116], and replicates were merged using samtools (v1.17). Fragment size distributions were calculated for merged replicates using deeptools bamPEFragmentSize (v3.5.1)[117]. The fragment size distributions shown in Supplementary Fig. 5a are smoothed for better visualisation by taking a running mean in 5 bp windows.

Peaks were called using MACS2 callpeak (v2.2.8)[118] with the parameters '--nomodel -shift -100 --extsize 200 -g mm' to call accessible regions for each replicate individually and for merged replicates. Consensus peak sets for $Pbk^{+/+}$ and $Pbk^{-/-}$ were created by taking peaks called in the merged replicates that were also called in at least 3 out of 4 individual replicates. Peaks and data on chromosomes 6 and 14 were excluded from downstream analysis because, by visual inspection, these chromosomes have globally reduced ATAC-seq coverage in $Pbk^{-/-}$.

**Differential accessibility.** For differential accessibility analysis, ATAC-seq read alignments were shifted to reflect the Tn5 cut site positions (+ 4 bp for plus strand, negative 5 bp for minus strand) and resized to 1 bp, using functions from the GenomicRanges R package (v1.46.1)[119]. Consensus accessible peaks from $Pbk^{+/+}$ and $Pbk^{-/-}$ were merged, and cut sites overlapping each merged peak in each replicate were counted. The counts were used as input for differential accessibility analysis with DESeq2 (v1.34.0)[120]. Shrunken log2 fold changes (using type = 'apeglm') were visualised using an MA plot (Fig. 5b), and the variance stabilising transformation was applied before performing principal component analysis (Supplementary Fig. 5b). Peaks with an adjusted *p*-value < 0.1 were defined as significantly different between $Pbk^{+/+}$ and $Pbk^{-/-}$.

Motif enrichment analysis in differentially accessible peaks was performed with AME (MEME suite v5.5.4)[121] using JASPAR 2024 CORE vertebrates non-redundant motifs[122] and merged consensus peaks from $Pbk^{+/+}$ and $Pbk^{-/-}$ as background. GO term enrichment analysis (Biological Process terms only) was carried out using clusterProfiler (v4.2.2)[123]. Genes with an annotated TSS overlapping a significantly differential peak were compared to genes with an annotated TSS overlapping any of the peaks used as input to the DESeq2 analysis.

**Footprinting analysis.** Transcription factor footprinting analysis was carried out using TOBIAS (v0.16.0)[124] with JASPAR 2024 CORE vertebrates non-redundant motifs[122]. Motifs in the top 5% by both p-value and absolute log2 fold change were selected as significant. Footprint plots in Supplementary Fig. 5 g show adjusted cut site density from TOBIAS in a ± 60 bp window around motif centres, with signal smoothed by taking the running mean in 5 bp windows to improve visualisation.

**Processing of published CTCF ChIP-seq and ATAC-seq data.** CTCF ChIP-seq data from preB cells from young mice[61] were downloaded from SRA (fasterq-dump, sratools v3.0.3) and aligned to GRCm39 using bowtie2 (v2.5.1)[125] in local mode and filtered using samtools (v1.16.1)[115] to retain only properly paired alignments with a quality score of ≥ 30. PCR duplicates were marked using sambamba (v1.0.1)[116]. Peaks were called using MACS2 (v2.2.8)[118]. Very few peaks were called for Rep2 therefore, only peaks from Rep1 were used for downstream analysis. CTCF motif locations were identified using the motifmatchr R package (v1.16.0, https://bioconductor.org/packages/motifmatchr) and the MA0139.2 motif from JASPAR2024[122]. Out of 10,424 CTCF peaks, 8532 had a CTCF motif match.

ATAC-seq data from preB cells from young mice[61] were downloaded from SRA (fasterq-dump, sratools v3.0.3) and trimmed using fastp (v0.23.3)[113] to remove adaptors and trailing bases with quality < 20. Reads were aligned to GRCm39 using bowtie2 (v2.5.1) in local

mode and filtered using samtools (v1.16.1) to retain only properly paired alignments with a quality score of ≥30. PCR duplicates were marked using sambamba (v1.0.1).

**Visualisation.** Heatmaps were plotted using deeptools (v3.5.1)[117]. Coverage tracks were created with deeptools bamCoverage (v3.5.1) with bin sizes of 10 bp or 1 bp and CPM normalisation. For ATAC-seq data, fragments of <100 bp were used to produce coverage tracks corresponding to accessible regions. Tracks at example loci were visualised using the IGV genome browser (v2.9.2)[126].

The nucleosome positioning plots in Fig. 5h were produced by calculating coverage of nucleosome-sized fragments (180–250 bp) around CTCF peaks (centred on CTCF motifs) using deeptools computeMatrix. Mean coverage was calculated in R and plotted using ggplot2. Flanking nucleosome positions were calculated by identifying signal maxima in a window covering 100–300 bp distance from the CTCF motif in both directions.

**CTCF ChIP-qPCR**
Mouse preB cells ($50$–$60 \times 10^6$) from asynchronous or mitotically arrested cultures were crosslinked in 10 ml 1% methanol-free formaldehyde (Thermo Scientific, 28906) in PBS (10 min, RT), quenched with glycine (Active Motif, 53008), 5 min, RT), washed once with PBS, snap-frozen and stored at negative 80 °C.

Mitotically arrested fixed cell pellets were thawed on ice, resuspended in 3–5 ml permeabilisation/blocking buffer (1X PBS, 10% normal goat serum, 2 mM EDTA, 0.1% Triton X-100, 1X cOmplete protease inhibitor cocktail (Roche, 11697498001)) and incubated at RT for 15 min. Samples were stained by the addition of RNase A (1 mg/ml final concentration), PI (0.05 mg/ml final concentration), and Alexa Fluor 488 anti-H3S10p antibody (1:1,000, Supplementary Table 2) and incubating at RT for 20 min, and then kept on ice until FACS. Mitotic cells were purified using a BD FACSAria Fusion flow cytometer with BD FACSDiva Software (v9.4) on the basis of high PI and Alexa Fluor 488 signals (4 N and H3S10p positive); FlowJo software (v10.8.1) was used to generate figures. Purified mitotic cells were pelleted, snap-frozen and stored at negative 80 °C.

Samples were processed using the ChIP-IT Express kit (Active Motif, 53008). Asynchronous cell pellets were resuspended in 3 ml ice-cold lysis buffer containing protease inhibitors (PIC) and PMSF, incubated on ice for 30 min and dounced 10 times. Nuclei were pelleted at $2400 \times g$ (10 min, 4 °C) and resuspended in shearing buffer containing PIC and PMSF at a concentration of $40 \times 10^6$ nuclei/ml. Purified mitotic cell pellets were resuspended directly in shearing buffer containing PIC and PMSF at a concentration of $1.4 \times 10^6$ cells per 200 µl; pellets of mitotic cells isolated on different days were pooled where necessary. To prepare sheared chromatin, 400 µl resuspended nuclei (corresponding to $16 \times 10^6$ cells) or 200 µl ($1.4 \times 10^6$) purified mitotic cells were transferred to 15 ml Bioruptor Tubes containing approximately 250 mg or 125 mg PBS-washed sonication beads (Diagenode, C010 20031), respectively. Samples were sonicated using a Bioruptor Pico (Diagenode) for 7 cycles of 30 s on and 30 s off, in 4 °C circulating water. Sheared chromatin was centrifuged at $18{,}000 \times g$ (10 min, 4 °C) and supernatant collected for chromatin immunoprecipitation (ChIP).

ChIP was performed according to the ChIP-IT Express kit recommendations (Active Motif), using 100 µl of sheared chromatin (corresponding to $4 \times 10^6$ asynchronous cells or $7 \times 10^5$ mitotic cells), 25 µl protein G magnetic beads and 3 µg or 1.4 µg of anti-CTCF antibody (Supplementary Table 2) in a 200 µl reaction incubated at 4 °C overnight. After washes and elution, ChIP and Input samples were reverse-crosslinked, treated with proteinase K and purified with the ChIP DNA Clean & Concentrator kit (Zymo, D5205). Quantitative real-time PCR was performed on Input (diluted to 1% equivalent) and ChIP samples using QuantiTect SYBR Green Master Mix (Qiagen, 204145), 10 µl reaction volume, primers listed in Supplementary Table 5) and a CFX96 Real-Time

System (Bio-Rad, CFX Manager v3.1). ChIP enrichment for each locus was calculated as a percentage of input; GraphPad Prism (v10.1.1) was used to prepare graphs and perform statistical comparisons. ChIP-qPCR was performed for three replicates per condition; one WT asynchronous replicate was excluded, where negative control regions showed very high signal compared to all other samples.

**Reporting summary**
Further information on research design is available in the Nature Portfolio Reporting Summary linked to this article.

## Data availability
Mass spectrometry proteomics data have been deposited to the ProteomeXchange Consortium via the PRIDE[127] partner repository with the dataset identifiers PXD057894 and PXD057898; processed data are provided in Supplementary Data 1 and 2. ATAC-seq data have been deposited to GEO with accession number GSE266554. Original image stacks have been uploaded to Figshare with [https://doi.org/10.6084/m9.figshare.29627411]. The UCSC genome browser guide RNA track BED file is available from https://hgdownload.soe.ucsc.edu/gbdb/mm10/crisprAll/crispr.bb. Published CTCF ChIP-seq data (Rep1: SRR6512735, SRR6512736; Rep2: SRR6512737, SRR6512738; Input: SRR6512739, SRR6512740) and ATAC-seq data (SRR6512723, SRR6512724, SRR6512725, SRR6512726) were downloaded from the NCBI SRA (SRP131401, corresponding to GEO accession GSE109671). JASPAR 2024 CORE vertebrate non-redundant data are available from https://jaspar.elixir.no/downloads/. GO annotations for use in Perseus were downloaded from http://annotations.perseus-framework.org (mainAnnot.mus_musculus.txt). The GO Ontology database used within PANTHER is identifiable with [https://doi.org/10.5281/zenodo.8436609]. Data from UniProt (https://www.uniprot.org/) and the MGI Gene Ontology Browser (https://www.informatics.jax.org/function.shtml) were used for additional manual annotations where indicated. Source data for Figs. 1b, d, 2a, c, e, f, 3c, d, 4b and Supplementary Figs. 4b, c, f, g and 5h, i are provided with this paper. All other data supporting the key findings of this study are available within the article and its supplementary files. Any further details and unique biological materials are available from the authors upon reasonable request.

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

## Acknowledgements

We thank the MRC LMS/National Institute for Health Research Imperial Biomedical Research Centre Flow Cytometry Facility; the MRC LMS Microscopy, Genomics, Proteomics and Bioinformatics facilities; and the MRC LMB Mass Spectrometry Facility for support. We thank Robert Hedley and Vasiliki Tsioligka for providing technical assistance in purifying mitotic cells at The Don Mason Facility of Flow Cytometry (Sir William Dunn School of Pathology, University of Oxford). We thank Shreya Jha, Nehir Nebioglu and George Young for assistance with ATAC-seq data processing and analysis. We thank Emilia Dimitrova and Tianyi Zhang for assistance with ChIP protocols. We thank Dorus Gadella, Ralf Kuehn, Stefan Stricker, Bradley Cobb and Stephen Smale for gifting reagents. This work was funded by a Kay Kendall Leukaemia Fund Junior Research Fellowship (KKL1334) awarded to A.D., and by support provided by the Medical Research Council UK (MC_UP_1605/12, MC_PC_23024, MC_PC_22015 awarded to A.G.F. and MC_UP_1605/11 awarded to M.M.). D.D. was supported by a Wellcome Trust Institutional Strategic Support Fund Springboard award (WCMA_PSN102). For the purpose of open access, the authors have applied a CC BY public copyright licence to any Author Accepted Manuscript version arising from this submission.

## Author contributions

A.D. and A.G.F. conceived the study, wrote the manuscript, and supervised the work. A.D. conducted most of the experiments, analysed data and produced the figures. D.H.G. engineered the PBK KO cell lines and performed additional experiments. E.I.-S. analysed the ATAC-seq data, C.W. assisted with microscopy and performed image analysis, B.P. isolated mitotic chromosomes by flow cytometry, and S.C. and K.B. performed additional experiments. H.B.K. and P.V.S. performed LC-MS/MS proteomics on chromosome and IP samples, respectively, and A.M. analysed these data. D.D. contributed to study design, assisted with experiments, and helped to identify ADNP as a bookmarking factor. M.M. provided input and guidance for the study and, as well as J.M.V., supported data analysis and the interpretation of results.

## Competing interests

The authors declare no competing interests.
