## [Transparent Peer Review file · Nature Communications]

PBK/TOPK mediates Ikaros, Aiolos and CTCF displacement from mitotic chromosomes and alters chromatin accessibility at selected C2H2-zinc finger protein binding sites

Corresponding Author: Professor Amanda Fisher

Version 0:

Reviewer comments:

Reviewer #1

(Remarks to the Author)

1. The localization of PBK substrates changes dramatically from prophase to anaphase. Therefore, whether cells were synchronized to the correct phase for analyses is critical to the conclusions, particularly in Figures 3-5. In the Methods section, it was stated that metaphase arrests are achieved through 0.1 $\mu\text{g/ml}$ demecolcine (Sigma-Aldrich, D1925) treatment for 5 hours. However, demecolcine typically arrests cells at prophase/prometaphase, not metaphase. Please clarify this point. Additionally, it would be helpful to specify if the cells for immunofluorescence (IF) were also synchronized using demecolcine. If not, it is necessary to validate the localization of PBK substrates in metaphase cells arrested by demecolcine.

2. Ikaros localizes to pericentric (heterochromatin) clusters in prophase and anaphase but is absent from metaphase chromosomes (Fig. c). It would be beneficial to demonstrate whether PBK is only expressed or active during metaphase in mitosis.

3. Please include the amino acid alignment of the linker region of these C2H2-ZF proteins in Figure 4c. Whether the less responsive proteins to Pbk^{-/-} have less conserved linkers. Additionally, it is unclear if Aurora or CDK1 also targets the linker regions of SP1 and YY1. Please clarify this point.

4. Both CTCF and Ikaros are PBK substrates. It is unclear why Ikaros re-associates with chromosomes in anaphase, while CTCF re-associates with chromosomes during telophase/cytokinesis. Please provide an explanation for this difference.

5. The statement that "CTCF retention on Pbk^{-/-} mitotic chromosomes leads to increased accessibility and altered nucleosome positioning" seems too strong, as Pbk^{-/-} affects many substrates, not just CTCF. Please consider revising this statement.

6. Page numbers should be added to the document for better reference.

Reviewer #2

(Remarks to the Author)

The manuscript by Dimond et al. describes a kinase (PBK/TOPK) that phosphorylates selected transcription factors to trigger their eviction from chromatin during mitosis. The authors used an elegant CRISPR strategy to knock in a fluorescent reporter (mNeonGreen) into the endogenous Ikaros locus to confirm that it is evicted from chromosomes during mitosis. They then show that PBK regulates mitotic chromosome compaction by phosphorylating multiple C2H2 Zinc finger TFs as well as CTCF. Retention of TFs on chromatin, due to PBK deletion, changes chromatin accessibility and nucleosome positioning,

presumably only on mitotic chromosomes, however it's not clear if this disruption persists during interphase.

Overall, this is a nice study with some interesting findings and observations. I have a few major comments and questions:

- It is interesting that PBK deleted cells are able to divide normally even though chromosome compaction is affected. Do the authors have an explanation for this? In other words, what is the consequence of retaining TFs at chromatin, how does that impact subsequent gene regulation during interphase, cell growth and phenotype? Some insight into this question would be good.

- Some ChIP-seq data is necessary for one of the factors, ideally CTCF to show how its binding is impacted at specific loci. Is CTCF occupancy impacted during interphase as a result of PBK loss or only during mitosis? Can the authors ChIP CTCF in mitotic cells? Cut and run could be used to avoid fixation and limiting cell numbers. Atac-seq is not sufficient.

- Why is atac-seq signal increased everywhere – is there a preference for some sites over others? What does global ATAC-seq look like? Not just over CTCF sites. This is important in order to determine the quality of the ATAC-seq data. The raw tracks for the replicates should be shown in Supplementary figures to understand the reproducibility.

- Why did CTCF not appear in the volcano plot in Figure 3E (C2H2-ZF)? Shouldn't it be enriched like the IKAROS proteins?

- The PBK KO data is convincing however this is long term depletion under steady state conditions, are the same effects seen with acute perturbation of PBK? Either with the inhibitor or some other method of rapidly perturbing the protein. This is to rule out indirect effects contributing to the observations. Does PBK KO affect CTCF protein levels for example? Similarly for the other TFs.

Version 1:

Reviewer comments:

Reviewer #1

(Remarks to the Author)

The authors have effectively addressed my concerns. The manuscript has been improved. I now consider it suitable for publication in Nature Communications. Here are some key highlights from the study:

1. The research validates that Ikaros dissociates from chromosomes during mitosis.
2. PBK has been identified as the primary kinase responsible for this dissociation.
3. Several factors, including CTCF, have been identified and confirmed as being evicted from chromosomes by PBK.
4. The study suggests that PBK-mediated phosphorylation and the subsequent displacement of C2H2-ZF factors may be crucial for proper chromosome compaction during mitosis.

Overall, this is a nice study.

Reviewer #2

(Remarks to the Author)

The authors provided thorough responses to each of my concerns. While most of the points have been addressed, we still have some remaining concerns.

The authors didn't address what the effects of retaining TFs at chromatin are on gene regulation/expression during interphase. Why wasn't RNA-seq performed to determine if there are any impacts on gene expression in the PBK KO cells?

Currently, the authors are inferring the effect of PBK on CTCF from the ATAC-seq data, without directly showing this at the genomic/chromatin level. ChIP-qPCR is not convincing enough, why wasn't it done using the PBK KO cells? The OTS514 inhibitor may not fully inhibit PBK. It would be much more useful to be able to quantify CTCF eviction using more reliable data e.g. ChIP-seq or Cut and Run.

Why couldn't the authors use flow purified mitotic chromosomes for Cut and Run? It is effective down to very low cell numbers.

Given that CTCF was not enriched in the proteomics volcano plot (Figure 3E) and no ChIP-seq was performed to show reduced CTCF occupancy during mitosis, it is hard to fully support the claims in the manuscript about CTCF eviction by PBK and its reinstatement in the PBK knockout context.

Reviewer #1 (Remarks to the Author):

1. The localization of PBK substrates changes dramatically from prophase to anaphase. Therefore, whether cells were synchronized to the correct phase for analyses is critical to the conclusions, particularly in Figures 3-5. In the Methods section, it was stated that metaphase arrests are achieved through 0.1 µg/ml demecolcine (Sigma-Aldrich, D1925) treatment for 5 hours. However, demecolcine typically arrests cells at prophase/prometaphase, not metaphase. Please clarify this point.

Cells were arrested with demecolcine (from Sigma-Aldrich), which according to the manufacturer's webpage arrests cells in metaphase. Demecolcine (also known as colcemid) acts by destabilising microtubules and thereby limiting spindle formation/kinetochore attachment and activating the SAC. Cells could therefore be regarded as being in a prometaphase-like state (Hayashi & Karlseder, 2013, doi.org/10.1038/onc.2012.615); however, the condensation of the chromosomes and the rest of the cell's status will more closely resemble metaphase (e.g. Menon *et al.*, 2021, doi.org/10.1016/B978-0-323-62520-3.00002-6). Indeed, most of the scientific literature refers to demecolcine as arresting cells in metaphase (for example, see Gribble *et al.*, 2009, doi.org/10.1038/nprot.2009.183; Ma & Poon, 2017, doi.org/10.1007/978-1-4939-6603-5_12; Moore & Best, 2001, doi.org/10.1038/npg.els.0001444) and standard 'metaphase spread' protocols generally arrest cells using demecolcine/colcemid. However, since we generally use the term 'mitotic arrest' (rather than metaphase arrest) we have changed the subtitle in the Methods section about addition of demecolcine to the same phrase (page 17) to avoid confusion.

Additionally, it would be helpful to specify if the cells for immunofluorescence (IF) were also synchronized using demecolcine. If not, it is necessary to validate the localization of PBK substrates in metaphase cells arrested by demecolcine.

Immunofluorescence was performed without demecolcine synchronisation to ensure that results were comparable with live-cell imaging (also without demecolcine) and were not an artefact of arrest. We have clarified the relevant figure legends by indicating that these experiments were performed on asynchronously dividing cultures (Figs. 1, 2, 4d and Supplementary Figs. 1, 2, 4i-j). We have also clarified which experiments use mitotic arrest (legends to Figs. 3a-d, 5a and Supplementary Figs. 3c, 4e).

We agree that it is an important control to validate localisation of PBK substrates in cells arrested by demecolcine. We have now performed the following experiments and included new figures in the revised manuscript:

1. Live-cell imaging of Ikaros-mNeonGreen localisation in demecolcine-arrested cells showing that Ikaros is re-localised to chromosomes following PBK inhibition with OTS514 (new Supplementary Fig. 2a, referred to in revised Results on page 6).
2. Immunofluorescence of Ikaros and CTCF in demecolcine arrested WT and PBK KO cells, showing chromosome localisation only in KO cells (new Supplementary Fig. 4e, referred to in revised Results on page 10).

2. Ikaros localizes to pericentric (heterochromatin) clusters in prophase and anaphase but is absent from metaphase chromosomes (Fig. c). It would be beneficial to demonstrate whether PBK is only expressed or active during metaphase in mitosis.

Based on prior literature, PBK/TOPK expression has been shown to generally increase in mitosis, and its activity is further regulated by phosphorylation (Matsumoto *et al.*, 2000; Gaudet, S *et al.*, 2000). Previous work has shown that the active, phosphorylated form of PBK appears in prophase and rapidly disappears at the transition to anaphase (Rizkallah *et al.*, 2015; figure provided below for reference). This appearance of active PBK in prophase (just ahead of Ikaros eviction), and disappearance in anaphase (just ahead of or coincident with Ikaros rebinding) correlates inversely with Ikaros localisation to chromosomes, consistent with the model we propose. To emphasise this interesting point, we have

referenced this in the revised Results (page 6) and Discussion (page 13). We thank the reviewer for raising this question and giving us the opportunity to clarify what was previously shown.

Provided for the reviewer: Panel taken from Figure 6b of Rizkallah *et al.*, *Oncotarget*, 2015, showing staining of active (phosphorylated) PBK/TOPK at different stages of mitosis.

[REDACTED]

3. Please include the amino acid alignment of the linker region of these C2H2-ZF proteins in Figure 4c. Whether the less responsive proteins to Pbk^{-/-} have less conserved linkers. Additionally, it is unclear if Aurora or CDK1 also targets the linker regions of SP1 and YY1. Please clarify this point.

We previously included an alignment of linker regions from key C2H2-ZF proteins (in Supplementary Fig. 6a) as part of the Discussion. We have now moved this alignment to Supplementary Fig. 4h and have added reference to this in the Results (page 10) and modified the Discussion accordingly (page 14). In our view there does not appear to be a clear correspondence between conservation of linker and PBK-responsiveness (for example SP1 shares an identical linker sequence with Ikaros but is unresponsive to PBK loss). Furthermore, our data suggest that all of these factors are phosphorylated at linkers in WT, but not in PBK KO cells. Rather, our hypothesis is that factors whose mitotic localisation is unresponsive to PBK loss may be additionally phosphorylated by other kinases at additional (different) sites. Aurora A and CDK1 are potential candidate kinases and target different sites to PBK (Aurora A sites are highlighted in the alignment). We have clarified this in the revised Results (page 10) and Discussion (pages 13-14).

4. Both CTCF and Ikaros are PBK substrates. It is unclear why Ikaros re-associates with chromosomes in anaphase, while CTCF re-associates with chromosomes during telophase/cytokinesis. Please provide an explanation for this difference.

The reviewer is correct in noticing that CTCF appears to re-associate with slightly slower kinetics than Ikaros during mitotic exit, and we have reconfirmed this observation (with an additional replicate

staining) and updated the corresponding legend to reflect this (Supplementary Fig. 4j in the revised manuscript).

Interestingly, we also observe a similar trend following OTS514 inhibitor treatment (as shown in new Supplementary Fig. 4i): Ikaros shows strong reassociation with <10 min OTS514 treatment, whereas CTCF reassociation was only weakly observed after 10 min, but was more strongly re-localised after 30 min of PBK inhibitor treatment. These observations are described in the revised Results (pages 10-11).

It is important to recognise that reassociation of factors will depend upon both the rate of dephosphorylation once PBK is inactivated, and on the rate of reassociation with chromatin. These rates will differ between factors, and could be influenced by the number of linkers, phosphatase recognition motifs, the chromatin state of binding locations, as well as a possibility that additional kinase regulators influence binding affinity. Our data suggest that dephosphorylation is likely mediated by PP1, which is known to be enriched at kinetochores during mitosis and colocalises with Ikaros at pericentromeric heterochromatin, and so it is possible this proximity/association contributes to the particularly rapid reassociation of Ikaros. We have commented on this difference and possible explanations to the revised Discussion (pages 13-14).

5. The statement that “CTCF retention on Pbk^{-/-} mitotic chromosomes leads to increased accessibility and altered nucleosome positioning” seems too strong, as Pbk^{-/-} affects many substrates, not just CTCF. Please consider revising this statement.

We agree that other factors could also be contributing and have revised this subheading (page 12) to “Increased accessibility and altered nucleosome positioning at CTCF binding sites in the absence of PBK”.

6. Page numbers should be added to the document for better reference.

We have added page numbers to the document.

Reviewer #2 (Remarks to the Author):

The manuscript by Dimond et al. describes a kinase (PBK/TOPK) that phosphorylates selected transcription factors to trigger their eviction from chromatin during mitosis. The authors used an elegant CRISPR strategy to knock in a fluorescent reporter (mNeonGreen) into the endogenous Ikaros locus to confirm that it is evicted from chromosomes during mitosis. They then show that PBK regulates mitotic chromosome compaction by phosphorylating multiple C2H2 Zinc finger TFs as well as CTCF. Retention of TFs on chromatin, due to PBK deletion, changes chromatin accessibility and nucleosome positioning, presumably only on mitotic chromosomes, however it's not clear if this disruption persists during interphase.

Overall, this is a nice study with some interesting findings and observations. I have a few major comments and questions:

We thank the reviewer for their positive comments and feedback which has helped to improve the manuscript and also suggested some important avenues for future investigations.

- It is interesting that PBK deleted cells are able to divide normally even though chromosome compaction is affected. Do the authors have an explanation for this? In other words, what is the consequence of retaining TFs at chromatin, how does that impact subsequent gene regulation during interphase, cell growth and phenotype? Some insight into this question would be good.

These are excellent questions, and it was surprising not to see further defects in PBK-deficient cells. Early indications are that PBK KO cells have slightly slower growth rates, and a slightly higher level of apoptosis can be observed in demecolcine-arrested cells (as indicated by a small subG1 peak seen in Supplementary Fig. 4b). However, these defects are mild, and the explanation/mechanism is still unclear. Whilst we agree these are interesting questions, as the phenotype is subtle, future follow-up studies may require *in vivo* work that is clearly beyond the scope of the current submission. For example, we note that some chromatin mutants that show altered chromosome compaction *in vitro*, such as DNA methylation, PRC2 and *Suv39h1/2* KO ESCs, do not show obvious mitotic defects in cultured cells (Djeghloul et al., 2020; Djeghloul et al., 2023). However, we know that *Suv39h1/h2* mice display aneuploidy and meiotic defects (Peters et al., Cell, 2001). Therefore, we believe that subtle mitotic defects observed *in vitro* may be far more significant in an *in vivo* context.

We have revised the text to incorporate some of these points in the Discussion (page 16).

- Some ChIP-seq data is necessary for one of the factors, ideally CTCF to show how its binding is impacted at specific loci. Is CTCF occupancy impacted during interphase as a result of PBK loss or only during mitosis? Can the authors ChIP CTCF in mitotic cells? Cut and run could be used to avoid fixation and limiting cell numbers. Atac-seq is not sufficient.

As an initial experiment, we performed ChIP-qPCR to check CTCF binding at specific loci (see figure below). In asynchronous preB cells we obtained good enrichment at known CTCF binding sites compared to negative control regions (up to ~100-fold enrichment), including CTCF peak/non-peak sites taken from Fig. 5d. In asynchronously dividing cells, we detected no difference in CTCF occupancy between WT and PBK KO cells (upper panel of figure). We repeated these experiments using mitotically arrested cells. Unfortunately, however, demecolcine treatment does not yield a pure metaphase population in mouse preB cells (<50%) with 'contaminating' interphase cells acting as a major confounder for this analysis (see Supplementary Fig. 4b), and variable responses between preB cell lines (e.g. appearance of a small subG1 population in arrested PBK KO cells). We therefore took a slightly different approach by comparing mitotically arrested WT cells treated with or without PBK inhibitor (thereby controlling for the proportion of contaminating interphase cells). As shown below (lower panels) there appears to be slightly increased CTCF occupancy following OTS514 treatment, especially when comparing matched treatments from the same experiment (dotted lines). Although this trend is

present for all CTCF peaks measured, the increase is only significant at two loci, and the magnitude of change is relatively small, potentially because mitotic differences are masked by contaminating interphase signal (note that enrichment is already high in untreated cells).

Figure for reviewer: CTCF ChIP-qPCR

CTCF ChIP-qPCR enrichment at known CTCF peaks, and negative control regions, as a percentage of input.

In asynchronous cells (upper panel), positive regions are up to ~100fold enriched over a negative control gene desert, and no differences are detected between WT and PBK KO cells.

In cells arrested with demecolcine (lower panel), enrichment at CTCF peaks remains high, likely reflecting contamination from interphase cells. Some increases are detected following 30 min of PBK inhibition with OTS514. Dotted lines link control/treated samples arising from the same arrested population; 2-way RM ANOVA, Dunnett's multiple comparisons test, **padj*<0.05, ****padj*<0.001.

Please note that for ATAC-seq analysis we were able to overcome the challenge of <50% arrest efficiency by performing the assay on flow-purified mitotic chromosomes. We acknowledge that future studies will undoubtedly require ChIP-seq or CUT&RUN on native isolated mitotic chromosomes, but analysis of this sort has not yet, to our knowledge, been performed by any parties. Although it is likely that CTCF retention mediates many of the observed alterations at CTCF loci, we cannot rule out that other factors (such as ADNP/ADNP2 which can bind the same motifs) contribute or are responsible. We have therefore modified the manuscript reflect these points (Results, page 12 (including subheading title) and Discussion, pages 14-15).

- Why is atac-seq signal increased everywhere – is there a preference for some sites over others? What does global ATAC-seq look like? Not just over CTCF sites. This is important in order to determine the quality of the ATAC-seq data. The raw tracks for the replicates should be shown in Supplementary figures to understand the reproducibility.

We apologise if this was not clear; however, ATAC-seq signal is not increased everywhere. As indicated in Fig. 5b, the majority of accessible sites across the genome (96,881 – 86%) do not change, and this is illustrated in Fig. 5d and Supplementary Fig. 5c (unchanging peaks indicated in grey vs significantly changing peaks in red). We have added percentages to Fig. 5b and modified the text (page 11) to clarify this point. Our motif enrichment analysis (Fig. 5e) revealed that there was a particular enrichment of CTCF motifs at sites showing increased accessibility which is why we then focused on CTCF sites in Figs. 5g and 5h.

As requested, we have now included the individual replicate tracks for the loci shown in Fig. 5d to demonstrate the reproducibility of the data. These are provided as new Supplementary Fig. 5d.

- Why did CTCF not appear in the volcano plot in Figure 3E (C2H2-ZF)? Shouldn't it be enriched like the IKAROS proteins?

We agree that this was surprising that CTCF was not enriched on the volcano plot in Fig. 3e. This could reflect a technical limitation of detection using LC-MS/MS or, more likely, that CTCF binding is in flux. We have acknowledged these possibilities more clearly in the revised text (page 10).

- The PBK KO data is convincing however this is long term depletion under steady state conditions, are the same effects seen with acute perturbation of PBK? Either with the inhibitor or some other method of rapidly perturbing the protein. This is to rule out indirect effects contributing to the observations. Does PBK KO affect CTCF protein levels for example? Similarly for the other TFs.

The reviewer makes an important point about whether long-term PBK depletion could cause indirect effects. We have already shown by western blot that PBK KO does not affect Ikaros levels (Figs. 2c & 2e), and have now repeated this analysis for CTCF, SP1 and YY1, included as new Supplementary Fig. 4f (referred to in Results on page 10). Furthermore, the input mass spec data for our IP experiment (Supplementary Fig. 4a and Supplementary Data 2) indicates that these factors are present at similar levels in mitotic lysates; we have plotted values for these factors separately to highlight this in new Supplementary Fig. 4g (referred to in Results on page 10).

Our results indicate that acute OTS514 treatment causes rapid widespread loss of linker phosphorylation (Supplementary Fig. 4c), similar to PBK KO cells which lack linker phosphorylation. We have now performed immunofluorescence staining for CTCF, SP1, YY1, alongside Ikaros, following OTS514 inhibitor treatment and included these data as new Supplementary Fig. 4i. Similar to the results in PBK KO cells, SP1 and YY1 remain un-associated in mitotic cells, whereas Ikaros and CTCF re-associate (albeit with slightly different dynamics) – this result is referred to in the revised Results on pages 10-11 and in the Discussion on page 14.

Reviewer #1 (Remarks to the Author):

The authors have effectively addressed my concerns. The manuscript has been improved. I now consider it suitable for publication in Nature Communications. Here are some key highlights from the study:

- 1. The research validates that Ikaros dissociates from chromosomes during mitosis.*
 - 2. PBK has been identified as the primary kinase responsible for this dissociation.*
 - 3. Several factors, including CTCF, have been identified and confirmed as being evicted from chromosomes by PBK.*
 - 4. The study suggests that PBK-mediated phosphorylation and the subsequent displacement of C2H2-ZF factors may be crucial for proper chromosome compaction during mitosis.*
- Overall, this is a nice study.*

We thank the reviewer for their comments and input which have helped to strengthen the conclusions of our manuscript.

Reviewer #2 (Remarks to the Author):

The authors provided thorough responses to each of my concerns. While most of the points have been addressed, we still have some remaining concerns.

The authors didn't address what the effects of retaining TFs at chromatin are on gene regulation/expression during interphase. Why wasn't RNA-seq performed to determine if there are any impacts on gene expression in the PBK KO cells?

This study is focused on mitosis and the mechanisms of evicting (or retaining) different transcription factors. There are many possible consequences of altered mitotic retention of factors; whilst the reviewer focuses on the idea of bookmarking and interphase gene expression, there could also be impacts on mitotic chromosome structure, genome integrity or mitotic progression, some of which may only manifest under certain conditions (e.g. stress or responses requiring changes to gene expression). Altogether, we believe that attempting to address the consequences of altered retention will require a dedicated follow-up study. In the context of possible gene expression differences, it may be most informative to study expression dynamics at high temporal resolution following mitotic exit; however, this will be technically challenging in preB cells, that are difficult to arrest/release in a synchronised manner.

Currently, the authors are inferring the effect of PBK on CTCF from the ATAC-seq data, without directly showing this at the genomic/chromatin level. ChIP-qPCR is not convincing enough, why wasn't it done using the PBK KO cells? The OTS514 inhibitor may not fully inhibit PBK. It would be much more useful to be able to quantify CTCF eviction using more reliable data e.g. ChIP-seq or Cut and Run.

As previously explained, demecolcine arrest in mouse preB cells is inefficient (<50% mitotic cells, see Supplementary Fig. 4b) and therefore 'contaminating' interphase cells are a major confounder for any downstream analysis.

However, in order to overcome this difficulty, and in line with the reviewer's suggestion to use PBK KO cells, we have now performed new experiments in which we specifically select/purify mitotic cells using flow cytometry, by sorting H3S10p positive cells. This has enabled us to perform CTCF ChIP-qPCR on asynchronous and pure mitotic populations from WT and PBK KO cells (new data shown in Supplementary Figs. 5h and 5i, and also included below). We detected strong enrichment at known CTCF binding sites in asynchronous populations, with no difference in CTCF occupancy between WT and PBK KO cells. In contrast, purified mitotic cells lacking PBK show significantly increased CTCF binding at these sites compared to WT controls (which show minimal evidence of mitotic CTCF binding). We have updated the Results (pg. 12), Discussion (pg. 15) and Methods (pgs. 30-35) sections accordingly, and annotated the peak locations tested from Fig 5d.

New Supplementary Figure 5h & 5i (see revised manuscript for full legends)

h) Flow cytometry gating strategy for purification of mitotic cells for ChIP based on PI and H3S10p staining.

i) CTCF ChIP-qPCR enrichment (as % of input) in asynchronous and purified mitotic *Pbk*^{+/+} and *Pbk*^{-/-} mouse preB cells, at known CTCF peaks and negative control sites.

We consider that robustly testing selected loci by ChIP-qPCR is sufficient at this stage to support our conclusions. Further genome-wide characterisation would be more appropriate for a future dedicated study where more extensive analysis and integration with other NGS experiments can be performed.

Why couldn't the authors use flow purified mitotic chromosomes for Cut and Run? It is effective down to very low cell numbers.

CUT&RUN on purified mitotic chromosomes is not straightforward since the standard protocol relies on a nuclear or cellular membrane to separate digested fragments from the rest of the genome, which is not possible with purified chromosomes. Whilst it may be possible to adapt and optimise the protocol in future, this is not a standard application of existing CUT&RUN protocols.

Given that CTCF was not enriched in the proteomics volcano plot (Figure 3E) and no ChIP-seq was performed to show reduced CTCF occupancy during mitosis, it is hard to fully support the claims in the manuscript about CTCF eviction by PBK and its reinstatement in the PBK knockout context.

We believe that the new CTCF ChIP data add significant strength to this claim, and we thank the reviewer for helping us to improve our manuscript. Altogether we have clearly shown that PBK is required for displacement of CTCF from chromosomes at a global level (immunofluorescence with PBK inhibitor or PBK KO shows global retention, Figs. 4d and S4e,i,j). Our new ChIP data

demonstrate that CTCF is specifically retained at interphase binding sites in mitotic PBK KO cells. This is supported by our mitotic ATAC-seq data showing increased accessibility, footprinting and altered nucleosome positioning at CTCF peak locations genome-wide in PBK KO conditions.

It is probably important to note that mitotic eviction of CTCF appears to be cell-type dependent, and therefore our claims are limited to mouse preB cells, used in this study. We have clarified this point in the revised Discussion (pg. 15).